# Sam68/KHDRBS1 is critical for colon tumorigenesis by regulating genotoxic stress-induced NF-κB activation

Kai Fu[1†], Xin Sun[1†], Eric M Wier[1], Andrea Hodgson[1,2], Yue Liu[1], Cynthia L Sears[2,3,4], Fengyi Wan[1,4*]

[1]Department of Biochemistry and Molecular Biology, Bloomberg School of Public Health, Johns Hopkins University, Baltimore, United States; [2]W. Harry Feinstone Department of Molecular Microbiology and Immunology, Bloomberg School of Public Health, John Hopkins University, Baltimore, United States; [3]Department of Medicine, Johns Hopkins University School of Medicine, Baltimore, United States; [4]Department of Oncology, Sidney Kimmel Comprehensive Cancer Center, Johns Hopkins Medical Institutions, Baltimore, United States

**Abstract** Nuclear factor kappa B (NF-κB)-mediated transcription is an important mediator for cellular responses to DNA damage. Genotoxic agents trigger a 'nuclear-to-cytoplasmic' NF-κB activation signaling pathway; however, the early nuclear signaling cascade linking DNA damage and NF-κB activation is poorly understood. Here we report that Src-associated-substrate-during-mitosis-of-68kDa/KH domain containing, RNA binding, signal transduction associated 1 (Sam68/KHDRBS1) is a key NF-κB regulator in genotoxic stress-initiated signaling pathway. Sam68 deficiency abolishes DNA damage-stimulated polymers of ADP-ribose (PAR) production and the PAR-dependent NF-κB transactivation of anti-apoptotic genes. Sam68 deleted cells are hypersensitive to genotoxicity caused by DNA damaging agents. Upregulated Sam68 coincides with elevated PAR production and NF-κB-mediated anti-apoptotic transcription in human and mouse colon cancer. Knockdown of Sam68 sensitizes human colon cancer cells to genotoxic stress-induced apoptosis and genetic deletion of Sam68 dampens colon tumor burden in mice. Together our data reveal a novel function of Sam68 in the genotoxic stress-initiated nuclear signaling, which is crucial for colon tumorigenesis.

*For correspondence: fwan1@jhu.edu

[†]These authors contributed equally to this work

Competing interests: The authors declare that no competing interests exist.

## Introduction

Nuclear factor kappa B (NF-κB) is a crucial transcription factor in a variety of pathophysiological conditions (*Dietz and Bahr, 2004*; *Grilli et al., 1993*; *Harhaj and Dixit, 2012*; *Hayden and Ghosh, 2008*; *Natoli, 2010*; *Smale, 2011*; *Vallabhapurapu and Karin, 2009*; *Wan and Lenardo, 2010*; *Wertz and Dixit, 2010*; *Wertz et al., 2004*). NF-κB responds to genotoxic threats (*e.g.* DNA damaging agents and γ-irradiation) via the activation of the inhibitor of NF-κB kinase (IKK) and NF-κB liberation from IκB proteins, similar to the canonical pathway activated by external stimuli (*Janssens et al., 2005*; *Perkins, 2007*; *Scheidereit, 2006*; *Wu and Miyamoto, 2007*). NF-κB signaling pathway has emerged as an important mediator for cellular responses to DNA damage, in particular NF-κB-conferred anti-apoptotic transcription facilitates the cell 'escape' from the lethal effects of DNA damage (*Janssens et al., 2005*; *Perkins, 2007*; *Scheidereit, 2006*; *Wu and Miyamoto, 2007*) and initiates cell cycle checkpoint control to promote cellular recovery from damage (*McCool and Miyamoto, 2012*; *Miyamoto, 2011*). Besides ataxia telangiectasia mutated (ATM) and IKKγ, two known crucial regulators of the genotoxic stress-activated NF-κB signaling pathway

**eLife digest** Cells use signaling pathways to detect and respond to harmful conditions by switching on genes that keep the cell healthy. One important pathway is the nuclear factor kappa B (NF-κB) signaling pathway, which is activated by many stimuli. These stimuli may come from infections from outside the cell or may originate inside the cell, as seen for DNA damage caused by irradiation, chemicals or rapid DNA replication in cancer cells.

Most of a cell's DNA is located in the cell nucleus. However, NF-κB proteins are normally located outside the nucleus, in the cell's cytoplasm. Damage to DNA triggers a signal from the nucleus to the cytoplasm. This signal activates the NF-κB proteins, which move into the nucleus and turn on genes that help the cell to recover from the damage. These genes include those that prevent the cell from self-destructing. In one step of the NF-κB activation process, chain-like molecules called polymers are made from a compound called poly(ADP-ribose), or PAR for short. However, few other details are known about how the damaged DNA in the nucleus signals to the cytoplasm.

A protein called Sam68, which is found in the cell nucleus, has been linked to DNA damage signaling. Fu, Sun et al. now present evidence that suggests that if mouse cells lack Sam68, they do not produce PAR polymers in response to DNA damage. In addition, these cells could not trigger the PAR-dependent signaling cascade that is essential for activating NF-κB and for turning on the protective genes. Consequently, cells that lacked Sam68 were extremely sensitive to agents that cause DNA damage, such as chemicals and irradiation.

The NF-κB pathway is regulated incorrectly in some cancers, but is also activated by DNA damage caused by cancer treatments. Therefore, Fu, Sun et al. also explored the role of Sam68 in cancer. Reducing the levels of Sam68 made human colon cancer cells more likely to self-destruct when they were exposed to DNA-damaging agents. Furthermore, removing Sam68 from mice that spontaneously grow colon cancer caused their tumors to develop more slowly than mice that retained Sam68 in their cells.

Overall, the findings presented by Fu, Sun et al. suggest that Sam68 regulates the signal from the nucleus to the cytoplasm that activates NF-κB proteins in response to DNA damage. Sam68 also appears to be important for helping colon cancer cells grow and survive. Future challenges will be to understand how Sam68 regulates the production of the PAR polymer in this response and to explore whether Sam68 can be targeted for treating cancer.

(*Li et al., 2001*; *Piret et al., 1999*), poly (ADP-ribose) polymerase 1 (PARP1) was recently revealed to be indispensable for the signaling cascade that links nuclear DNA damage recognition to cytoplasmic IKK activation (*Stilmann et al., 2009*). Sequential post-translational modifications, including phosphorylation, ubiquitination and SUMOylation, of these signaling regulators are critical for NF-κB activation following DNA damage (*Huang et al., 2003*; *Mabb et al., 2006*; *Wu et al., 2006*), in particular, PARP1-catalyzed poly (ADP-ribosyl)ation (PARylation) has emerged as a vital means for rapid assembly of the signaling complexes that are critical for DNA damage-initiated NF-κB activation (*Mabb et al., 2006*; *Stilmann et al., 2009*). Although these studies have considerably advanced our understanding of the cellular response to DNA damage, the genotoxic stress-initiated ''nuclear-to-cytoplasmic'' NF-κB signaling pathway remains poorly understood, in particular the early signaling networks linking DNA lesion recognition in the nucleus to subsequent activation of IKK and liberation of NF-κB in the cytoplasm.

Sam68 (Src-associated substrate during mitosis of 68 kDa, also named KH domain containing, RNA binding, signal transduction associated 1 [KHDRBS1], and encoded by *KHDRBS1* gene), an RNA-binding protein that preferentially resides in the nucleus, plays versatile functions in an increasing number of cellular processes (*Bielli et al., 2011*; *Cheung et al., 2007*; *Fu et al., 2013*; *Glisovic et al., 2008*; *Henao-Mejia et al., 2009*; *Huot et al., 2012*; *Iijima et al., 2011*; *Lukong and Richard, 2003*; *Matter et al., 2002*; *Paronetto et al., 2009*; *Rajan et al., 2008a*, *2008b*; *Ramakrishnan and Baltimore, 2011*; *Richard, 2010*; *Sette, 2010*; *Yang et al., 2002*). Through its KH (heteronuclear ribonucleoprotein particle K homology) domain, Sam68 is capable of binding single- and double-stranded DNA in addition to RNA (*Lukong and Richard, 2003*). Of note,

Sam68 was identified as a PAR-binding protein in alkylating agent treated cells (*Gagne et al., 2008*) and a putative substrate of ATM, ATM and Rad3-related (ATR), and DNA-dependent protein kinase (DNA-PK) (*Beli et al., 2012*), which suggests that Sam68 could be an important molecule in the cellular response to DNA damage. Although emerging evidence suggests the involvement of Sam68 in multiple signaling pathways, it has not been extensively investigated yet whether Sam68, an almost strictly nuclear protein, participates in the signal communication network of nuclear-initiated signaling pathways. Moreover, aberrant expression of Sam68 has been acknowledged in multiple cancers and elevated Sam68 expression correlates with tumor progression and poor prognosis in cancer patients (*Chen et al., 2012*; *Liao et al., 2013*; *Song et al., 2010*; *Zhang et al., 2009*). Overexpression of Sam68 has been proposed as a prognostic marker (*Chen et al., 2012*; *Liao et al., 2013*; *Song et al., 2010*; *Zhang et al., 2009*), however, the precise function of Sam68 in cancer development and survival remains obscure.

Here we report that Sam68 is an important regulator in genotoxic stress-initiated early signaling in the nucleus, which leads to NF-κB activation. Sam68 deletion diminishes DNA damage-stimulated PARP1 activation and PAR production, as well as the PAR-dependent NF-κB signaling and transactivation of an array of anti-apoptotic genes. As a consequence, Sam68 knockout cells are hypersensitive to genotoxicity caused by γ-irradiation and DNA damaging chemicals. Moreover, downregulation of Sam68 substantially sensitizes human colorectal cancer cells to spontaneous and genotoxic stress-induced cell death and retards colon tumor growth and survival in genetically susceptible *Apc*^min716/+ mice. Hence our data reveal a crucial function of Sam68 in the genotoxic stress-initiated 'nuclear to cytoplasmic' NF-κB transactivation and the involvement of Sam68 in the development and survival of colon cancer.

## Results

### Sam68 is essential for DNA damage-induced NF-κB activation

To examine the potential role of Sam68 in nuclear-initiated NF-κB activation, we first compared the genotoxic stress-induced NF-κB signaling in immortalized wild-type and Sam68 knockout (KO) mouse embryonic fibroblasts (MEFs). As expected, Camptothecin (CPT), a DNA-damaging chemical that inhibits DNA topoisomerase I (*Stilmann et al., 2009*), stimulated a rapid degradation of IκBα, a prerequisite for NF-κB liberation and transactivation, in a dose- and time-dependent manner in wild-type MEFs (*Figure 1A–B* and *Figure 1—figure supplement 1A*). In contrast, DNA damage-induced IκBα degradation was remarkably attenuated in Sam68 KO MEFs (*Figure 1A–B*) and MEFs without poly(ADP-ribose)polymerase 1 (PARP1) (*Figure 1—figure supplement 1B*), a recently identified key nuclear regulator of DNA damage-induced NF-κB activation (*Stilmann et al., 2009*). Moreover, CPT treatment triggered remarkable nuclear translocation of NF-κB in wild-type MEFs (*Figure 1C*); additionally, the derived nuclear extracts formed high-affinity binding complexes with immunoglobin (Ig) κB double-stranded DNA, which was further confirmed by cold oligonucleotide competition and super shift assays (*Figure 1D*). In striking contrast, NF-κB nuclear accumulation and binding capacity to Ig κB DNA were almost abolished in the CPT-stimulated Sam68 KO MEFs (*Figure 1C–D*). To ascertain that Sam68 deficiency solely results in the impaired genotoxic stress-induced NF-κB activation in MEFs, we examined the NF-κB signaling in wild-type MEFs with Sam68 knockdown by small interference RNAs (siRNAs) and Sam68 KO MEFs supplemented with exogenous Sam68. CPT-triggered IκBα degradation was substantially reduced in Sam68-specific siRNA-expressing wild-type MEFs, compared to scrambled non-specific siRNA-transfected cells (*Figure 1—figure supplement 1C*), whereas ectopic expression of green fluorescent protein (GFP) tagged Sam68, but not GFP alone, markedly restored CPT-induced IκBα degradation in Sam68 KO MEFs (*Figure 1—figure supplement 1D–E*). Together, these results suggest that Sam68 could execute an essential function in the nuclear-initiated NF-κB signaling pathway.

### Sam68 participates in DNA damage-initiated early NF-κB signaling in the nucleus

Sam68 is an almost strictly nuclear protein; its nuclear import is conferred by a nuclear localization signal (NLS) in the C-terminus (*Fu et al., 2013*; *Ishidate et al., 1997*; *Lukong and Richard, 2003*). Consistently, the ΔC (deletion of C-terminal NLS) mutant of Sam68 switches its preferred subcellular

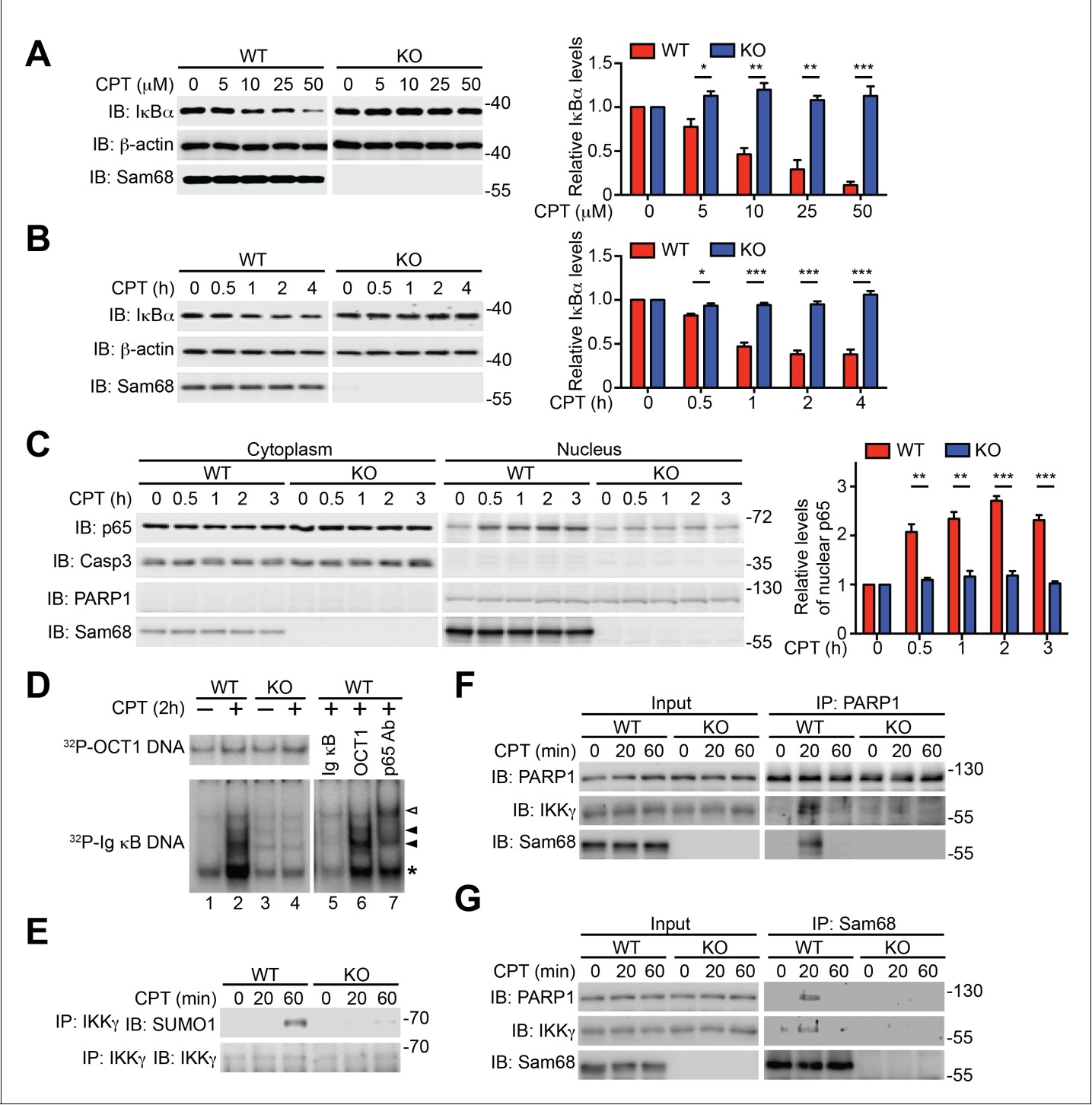

**Figure 1.** Sam68 is required for the nuclear-initiated NF-κB signaling in response to DNA damage. (**A** and **B**) Whole cell lysates from wild-type (WT) and Sam68 knockout (KO) mouse embryonic fibroblasts (MEFs) treated with indicated concentrations of CPT for 2 hr (**A**) or 10 μM of CPT for indicated periods (**B**) were immunoblotted (IB) for IκBα and Sam68, with β-actin as a loading control. *Right*, the IκBα levels, normalized to β-actin and untreated controls, were quantified from three independent experiments. (**C**) Cytosolic and nuclear fractions derived from WT and Sam68 KO MEFs stimulated with 10 μM of CPT for indicated periods were IB for indicated proteins. Caspase-3 (Casp3) and PARP1 served as loading controls and cytosolic and nuclear markers, respectively. *Right*, the p65 levels in the nucleus, normalized to PARP1 and untreated controls, were quantified from three independent experiments. (**D**) Nuclear extracts of WT and Sam68 KO MEFs treated with (+) or without (−) CPT (10 μM, 2 hr) were analyzed by EMSA with ³²P-labeled immunoglobin (Ig) κB or OCT1 oligonucleotides. In some cases, EMSA was performed in the presence of 100-fold unlabeled Ig κB or OCT1 oligonucleotide competitors (lanes 5–6) or p65 antibody (Ab) (lane 7). Ig κB DNA binding complexes are labeled with filled triangles, and the

*Figure 1 continued on next page*

*Figure 1 continued*

supershifted band and nonspecific band are labeled with an open triangle and an asterisk, respectively. (E) WT and Sam68 KO MEFs were stimulated with 10 µM of CPT for indicated periods, and immunoprecipitants (IP) with IKKγ antibody were immunoblotted for indicated proteins. (F and G) Whole cell lysates (Input) from WT and Sam68 KO MEFs stimulated with 10 µM of CPT for indicated periods were IB directly or after IP with PARP1 antibody (F) or Sam68 antibody (G) for indicated proteins. Data are representative of at least three independent experiments. Results in (A, B and C) are expressed as mean and s.e.m. *p<0.05, **p<0.01, ***p<0.001 by Student's *t* tests.

The following figure supplements are available for figure 1:

**Figure supplement 1.** Sam68 is critical for genotoxic stress-induced IκBα degradation.

**Figure supplement 2.** Sam68 complexes with PARP1 and IKKγ during the cellular response to genotoxic stress.

localization to the cytoplasm, in contrast to the strict nuclear accumulation observed in wild-type Sam68 (*Figure 1—figure supplement 1D*). Sam68 KO MEFs reconstituted with wild-type Sam68, but not GFP control, displayed restored CPT-triggered NF-κB activation signaling including IκBα degradation, phosphorylation of p65, and p65 nuclear accumulation (*Figure 1—figure supplement 1E–F*). In contrast, supplementing with the Sam68 (ΔC) mutant failed to restore CPT-triggered IκBα degradation (*Figure 1—figure supplement 1E*), which indicates an indispensable role of the Sam68 nuclear function in DNA damage-initiated NF-κB activation. Indeed, CPT treatment-induced SUMOylation of IKKγ, a pivotal post-translational modification on IKKγ in the nucleus that subsequently leads to activation of IKKβ and liberation of NF-κB in the cytoplasm (*Huang et al., 2003*; *Mabb et al., 2006*), was substantially attenuated in Sam68 KO MEFs in comparison to wild-type controls (*Figure 1E*). Moreover, assembly of the PARP1-IKKγ signal complex has been established as a prerequisite for cytoplasmic NF-κB activation in the cellular response to genotoxic stresses (*Gibson and Kraus, 2012*; *McCool and Miyamoto, 2012*; *Miyamoto, 2011*). As expected, PARP1 complexed with IKKγ following CPT treatment, whereas there was no detectable PARP1-IKKγ interaction in the absence of genotoxic stress in wild-type MEFs (*Figure 1F*). In striking contrast, the CPT-induced PARP1-IKKγ interaction was abolished in Sam68 KO MEFs (*Figure 1F*). Moreover, strong inducible interactions among Sam68, PARP1, and IKKγ were observed in wild-type MEFs, at 20 min after CPT treatment (*Figure 1F–G*). Of note, immune-depletion of Sam68 using antibodies abolished the CPT-induced PARP1-IKKγ interaction in wild-type MEFs (*Figure 1—figure supplement 2*), suggesting that PARP1 interacts with Sam68 and IKKγ simultaneously. Hence our results suggest that Sam68 participates in the early nuclear signaling cascade in DNA damage-initiated NF-κB activation.

## Sam68 controls DNA damage-induced PARylation

Beyond its indispensable role in DNA repair (*Gibson and Kraus, 2012*), emerging evidence demonstrates that PARP1-mediated PARylation is one of the most crucial post-translational modifications orchestrating DNA damage-initiated NF-κB signaling (*Stilmann et al., 2009*). The inducible Sam68-PARP1 interaction following genotoxic stress led us to examine whether Sam68 impacts DNA break-stimulated PAR synthesis. As expected, CPT treatment induced a wide array of PARylated proteins peaking at 20 min post stimulation in wild-type MEFs, which were diminished by one-hour pretreatment with two independent PARP inhibitors, Olaparib and PJ-34, thus confirming the PAR specificity (*Figure 2A* and *Figure 2—figure supplement 1A*). Strikingly, genotoxic stress-induced PAR synthesis was markedly dampened in Sam68 KO MEFs (*Figure 2A* and *Figure 2—figure supplement 1A*). Consistently, genotoxic stress-induced PARylation of the known PARP1 substrates including PARP1 itself, NBS1, and Ku70 was also substantially attenuated in Sam68 KO MEFs (*Figure 2—figure supplement 1B*). PAR that is synthesized after DNA damage can be rapidly degraded by PAR glycohydrolase (PARG), therefore we examined the possibility that PARG is hyper-activated in the absence of Sam68, thus degrading the formed PAR chains immediately after genotoxic stress. However, down-regulation of PARG by siRNA in Sam68 KO MEFs failed to restore CPT-induced PAR production to a similar level to that in wild-type MEFs (*Figure 2—figure supplement 2*), which suggests that the attenuated PAR formation in the absence of Sam68 is not caused by a rapid degradation of PAR chains. Furthermore, CPT and γ-irradiation stimulated PAR production was markedly reduced in

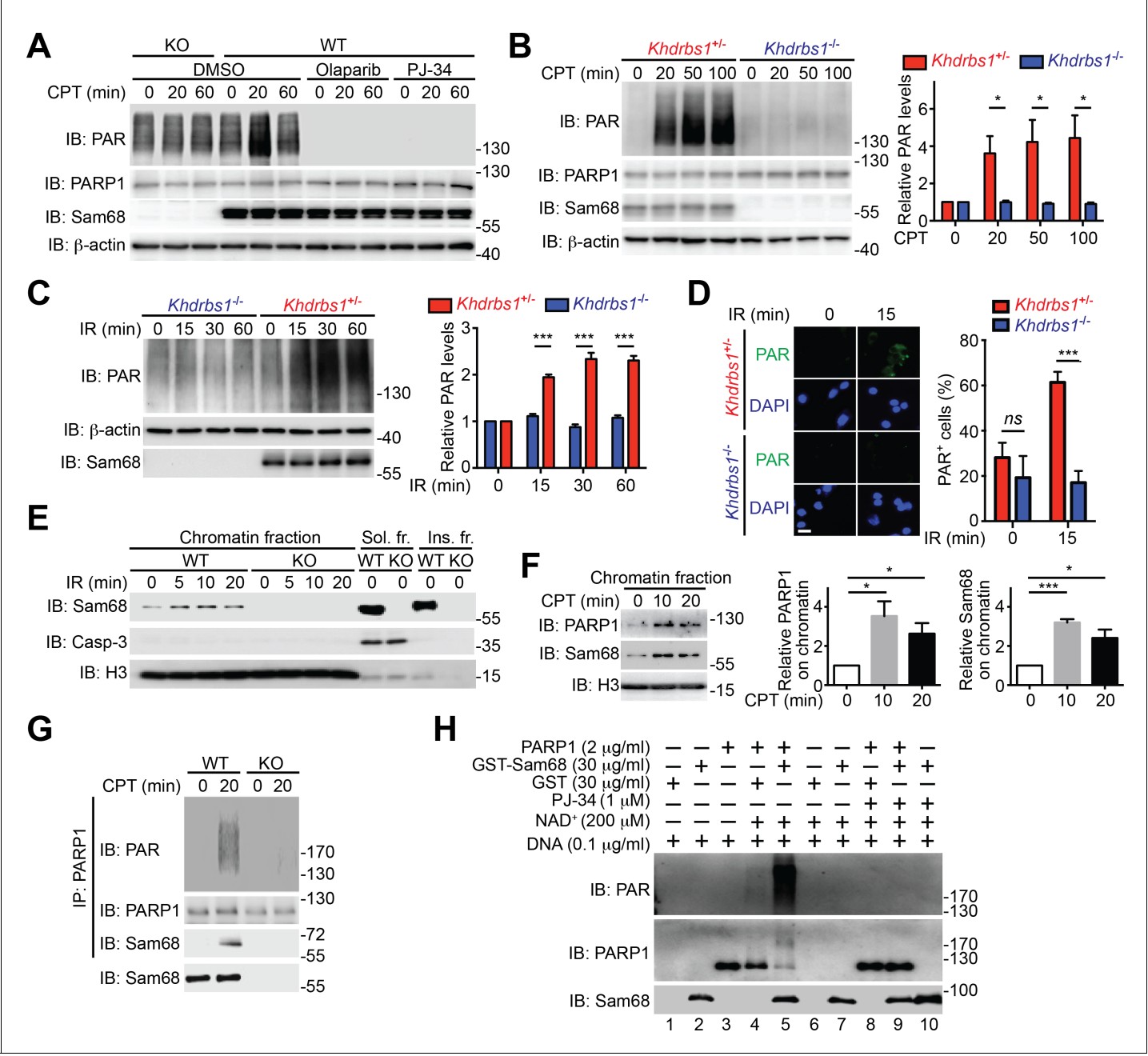

**Figure 2.** Sam68 facilitates PARP1-catalyzed PARylation in response to DNA damage. (**A**) Wild-type (WT) and Sam68 knockout (KO) mouse embryonic fibroblasts (MEFs) pretreated with Olaparib (10 μM), PJ-34 (10 μM), or DMSO for 1 hr, were stimulated with 10 μM of CPT for indicated periods, and whole cell lysates were immunoblotted (IB) for indicated proteins, with β-actin as a loading control. (**B** and **C**) Primary colonic epithelial cells (CECs) isolated from *Khdrbs1*[+/-] and *Khdrbs1*[-/-] mice were stimulated with 10 μM of CPT (**B**) or 10 Gy of γ-irradiation (IR) (**C**). Whole cell lysates were derived at indicated periods post stimulation and IB as in (**A**). *Right*, the PAR levels, normalized to β-actin and untreated controls, were quantified from three independent experiments. (**D**) Immunofluorescence micrographs of PARylated proteins (PAR) in CECs treated as in (**C**), with nuclei counterstained by DAPI. Scale bar, 10 μm. Percentage of CECs (>100 cells from 5–8 random fields) with PAR staining was quantified (right). (**E**) WT and Sam68 KO MEFs were γ-irradiated (IR) at 10 Gy and the chromatin, soluble (Sol. fr.), and insoluble (Ins. fr.) subcellular fractions were derived at indicated time points post γ-irradiation and IB for Sam68, Caspase-3 (Casp-3), and Histone H3 (H3). (**F**) WT MEFs were stimulated with 10 μM of CPT for indicated periods and the chromatin fractions were derived and IB as in (**E**). *Right*, the Sam68 and PARP1 levels in the chromatin fractions, normalized to H3 and untreated controls, were quantified from three independent experiments. (**G**) Whole cell lysates from WT and Sam68 KO MEFs stimulated with 10 μM of CPT for 20 min, were IB directly or after immunoprecipitation (IP) with PARP1 antibody for indicated proteins. (**H**) Recombinant PARP1 protein was incubated in reaction buffer containing damaged DNA or with purified GST or GST-Sam68 protein in the presence or absence of NAD[+] or the PARP1 inhibitor PJ-

*Figure 2 continued on next page*

*Figure 2 continued*

34, as indicated. The reaction mixture was separated by SDS-PAGE and subjected to IB with the PAR, PARP1, and Sam68 antibodies. Results in (**B**, **C**, **D** and **F**) are expressed as mean and s.e.m. ns, non-significant difference and , *p<0.05, ***p<0.001 by Student's *t* tests. Data are representative of at least three independent experiments.

The following figure supplements are available for figure 2:

**Figure supplement 1.** Sam68 deletion attenuates genotoxic stress-stimulated PARylation.

**Figure supplement 2.** Down-regulation of PARG does not restore DNA damage-induced PAR production in Sam68 KO MEFs.

**Figure supplement 3.** Sam68 functions upstream of PARP1 in cellular response to genotoxic stresses.

**Figure supplement 4.** Sam68-stimulated PARP1 PARylation is DNA dependent.

primary colonic epithelial cells (CECs) derived from *Khdrbs1*$^{-/-}$ (Sam68 knockout) mice, compared to *Khdrbs1*$^{+/-}$ (Sam68 heterozygote) controls (*Figure 2B–D*), thus supporting the critical function of Sam68 in controlling DNA damage-stimulated PAR production.

Given the evidence that PARP1 is rapidly recruited to DNA damage sites (*Krishnakumar and Kraus, 2010*), we performed chromatin fractionation assays to examine the possibility that Sam68 can be recruited to DNA lesions. Remarkably, Sam68 was enriched in chromatin fractions in MEFs following γ-irradiation (*Figure 2E*) and CPT treatment (*Figure 2F*), and the kinetics of Sam68 enrichment was similar to that of PARP1 on damaged chromatin (*Figure 2F*), which supports the inducible interaction between Sam68 and PARP1 in the early DNA damage signaling (*Figure 1F–G*). Moreover, Sam68 and PARP1 were still substantially enriched in chromatin fractions in the PARP1-inhibited MEFs following γ-irradiation (*Figure 2—figure supplement 3A*), indicating that genotoxic stress-induced Sam68 enrichment in chromatin is not dependent on PAR formation. We further examined whether ectopic expression of PARP1 could rescue the defect in DNA damage-induced PAR formation caused by Sam68 loss. As expected, supplementing exogenous GFP-Sam68, compared to the GFP control, largely restored γ-irradiation-stimulated PAR production in Sam68 KO MEFs (*Figure 2—figure supplement 3B*). In contrast, PARP1 overexpression did not rescue the PAR chain formation in response to γ-irradiation under a Sam68 deleted condition (*Figure 2—figure supplement 3B*), which suggests that Sam68 is required for DNA damage-triggered PARP1 activation and is consistent with Sam68 being an essential upstream activator of PARP1.

DNA-dependent PARP enzymes, *i.e.* PARP1 and PARP2, are activated during the cellular response to DNA damage, of which PARP1 is the most robust enzyme that catalyzes >90% of PAR formation in cells (*Krishnakumar and Kraus, 2010*). In contrast to the genotoxic stress-induced Sam68-PARP1 interaction (*Figure 1F–G* and *Figure 1—figure supplement 2D*), no detectable interaction between Sam68 and PARP2 was observed under either unstimulated or damaged conditions (*Figure 1—figure supplement 2D*). Provided the crucial funciton of PARP1 in DNA damage responses, we therefore examined whether Sam68 impacts PARP1-conferred PARylation following DNA damage. In line with the vigorous PAR production in the presence of genotoxic stress (*Figure 2A*), immunoprecipitated PARP1 was associated with Sam68 and various PARylated target proteins in CPT-treated wild-type MEFs (*Figure 2G*). In contrast, the PARylated species were dramatically reduced despite an equal amount of PARP1 immunoprecipitated from Sam68 KO MEFs was used (*Figure 2G*). Conversely, supplementing with GFP-tagged Sam68, in comparison to a GFP control, markedly augmented the DNA damage-triggered PAR synthesis in CPT-treated Sam68 KO MEFs (*Figure 2—figure supplement 3B*), further supporting the key function of Sam68 in controlling DNA damage-initiated PARylation. To assess the direct impact of Sam68 on PARP1-catalyzed PAR production, we utilized recombinant PARP1 and Sam68 proteins in in vitro PARylation assays. DNA damage-activated PARP1 auto-modified itself with the addition of PAR moieties in the presence of nicotinamide adenine dinucleotide (NAD$^+$) and DNA, as indicated by the formation of PARylated PARP1 species (*Figure 2H* and *Figure 2—figure supplement 4A*) and a corresponding reduction in unmodified PARP1 protein (*Figure 2H* and *Figure 2—figure supplement 4B*, compare lane 4 with lane 3). Strikingly, incubation of recombinant Sam68 protein, but not GST control, with PARP1

dramatically boosted PAR production, which was paralleled by a sharp loss of unmodified PARP1 protein (*Figure 2H* and *Figure 2—figure supplement 4B*, compare lane 5 with lane 4). Of note, the amount of Sam68 protein from the boosted PARylation reaction remained at a comparable level (*Figure 2H*, compare lane 5 with lane 7), thus ruling out the possibility that Sam68 is a substrate of PARP1 in vitro and indicating that Sam68 harbors a stimulatory function for DNA-dependent PARP1 activity. In the absence of PARP1, neither GST nor GST-Sam68 protein exhibited PARylation activity (*Figure 2H*, lanes 1, 2, 6, 7, and 10), indicating that Sam68 per se does not possess the enzymatic activity to transfer ADP-ribosyl polymers. Moreover, in the absence of damaged DNA, incubation of Sam68 and PARP1 failed to form detectable PAR chains from supplemented NAD$^+$ (*Figure 2—figure supplement 4C*), which suggests that Sam68 stimulates DNA-dependent PARP1 activation and subsequent PAR production.

## The Sam68-PARP1 interaction is critical for genotoxic stress-induced PARylation and NF-κB activation

Using various Sam68 truncates (*Figure 3A*), we sought to understand the key domain(s) in Sam68 needed for its interaction with PARP1. We detected the association of the full-length, ΔC, and ΔKH truncated Sam68 to PARP1, but not GFP control (*Figure 3B* and *Figure 3—figure supplement 1A*). In contrast, deletion of the N-terminal amino acids 1–102 (ΔN) of Sam68 almost abolished the association of Sam68 to PARP1 (*Figure 3B* and *Figure 3—figure supplement 1A*), suggesting that the N--terminal residues are critical for the Sam68-PARP1 interaction. Moreover, our pull-down assays using recombinant proteins demonstrated a direct Sam68-PARP1 interaction (*Figure 3C*). In contrast to the strong association between PARP1 and full-length Sam68, ΔN truncated Sam68 protein barely interacted with PARP1 (*Figure 3C*), which further supports the critical role of N-terminus of Sam68 for the Sam68-PARP1 interaction. To examine the functional importance of the Sam68-PARP1 interaction, we compared DNA damage-stimulated NF-κB signaling in Sam68 KO MEFs supplemented with full-length or ΔN mutant Sam68. Transient transfection of full-length Sam68, but not GFP control, significantly restored genotoxic stress-induced PARylation (*Figure 3D* and *Figure 3—figure supplement 1C*), assembly of the PARP1-IKKγ signaling complex (*Figure 3D*), cytoplasmic degradation of IκBα, and nuclear translocation of p65 (*Figure 3—figure supplement 1C*), consistent with our previous observation (*Figure 1F–G*, *2G*, and *Figure 1—figure supplement 1E–F*). In contrast, despite full-length Sam68 and Sam68 (ΔN) truncate sharing strict nuclear localization (*Figure 3E*), ectopic expression of Sam68 (ΔN) mutant failed to restore the DNA damage-triggered PARylation, PARP1-IKKγ signal complex assembly, IκBα degradation, and p65 nuclear accumulation (*Figure 3D* and *Figure 3—figure supplement 1B–C*). Moreover, we carried out in vitro PARylation assays using recombinant Sam68 and Sam68 (ΔN) proteins, to examine the role of the Sam68-PARP1 interaction for Sam68-stimulated PARP1 activation. Consistent with our previous observation (*Figure 2H*), incubation of Sam68 with PARP1 substantially enhanced PAR formation in the presence of damaged DNA and NAD$^+$ (*Figure 3—figure supplement 1D*, compare lane 3 with lane 7). However, the stimulatory effect of Sam68 on PARP1 activation and PARylation under the same condition was dramatically impeded by Sam68 (ΔN) protein (*Figure 3—figure supplement 1D*, compare lane 7 with lane 11), suggesting that the Sam68-PARP1 interaction is essential for Sam68 to stimulate PARP1 activation. Therefore our results demonstrate that the N-terminus of Sam68 is important for the association between Sam68 and PARP1 and the Sam68-PARP1 interaction is critical for DNA damage-induced PARylation and PAR-dependent NF-κB signaling.

## Sam68 deletion impairs genotoxic stress-induced NF-κB signaling in mouse primary cells

To explore the relevance of Sam68 in primary cells, we examined the DNA damage-initiated NF-κB activation in CECs, known for their DNA damage-hypersensitivity, from *Khdrbs1*$^{+/-}$ and *Khdrbs1*$^{-/-}$ mice. CPT- and γ-irradiation-induced IκBα degradation rapidly occurred in *Khdrbs1*$^{+/-}$ CECs, but was significantly attenuated in *Khdrbs1*$^{-/-}$ cells (*Figure 4A* and *Figure 4—figure supplement 1A*). In parallel, γ-irradiation and CPT triggered p65 nuclear translocation in *Khdrbs1*$^{+/-}$ CECs, whereas the nuclear accumulation of p65 was nearly abolished in *Khdrbs1*$^{-/-}$ cells (*Figure 4B–C* and *Figure 4—figure supplement 1B*). Moreover, in *Khdrbs1*$^{+/-}$ CECs, γ-irradiation stimulated the SUMOylation of IKKγ and migration of IKKγ from the nucleus to the cytoplasm, both required signaling events for

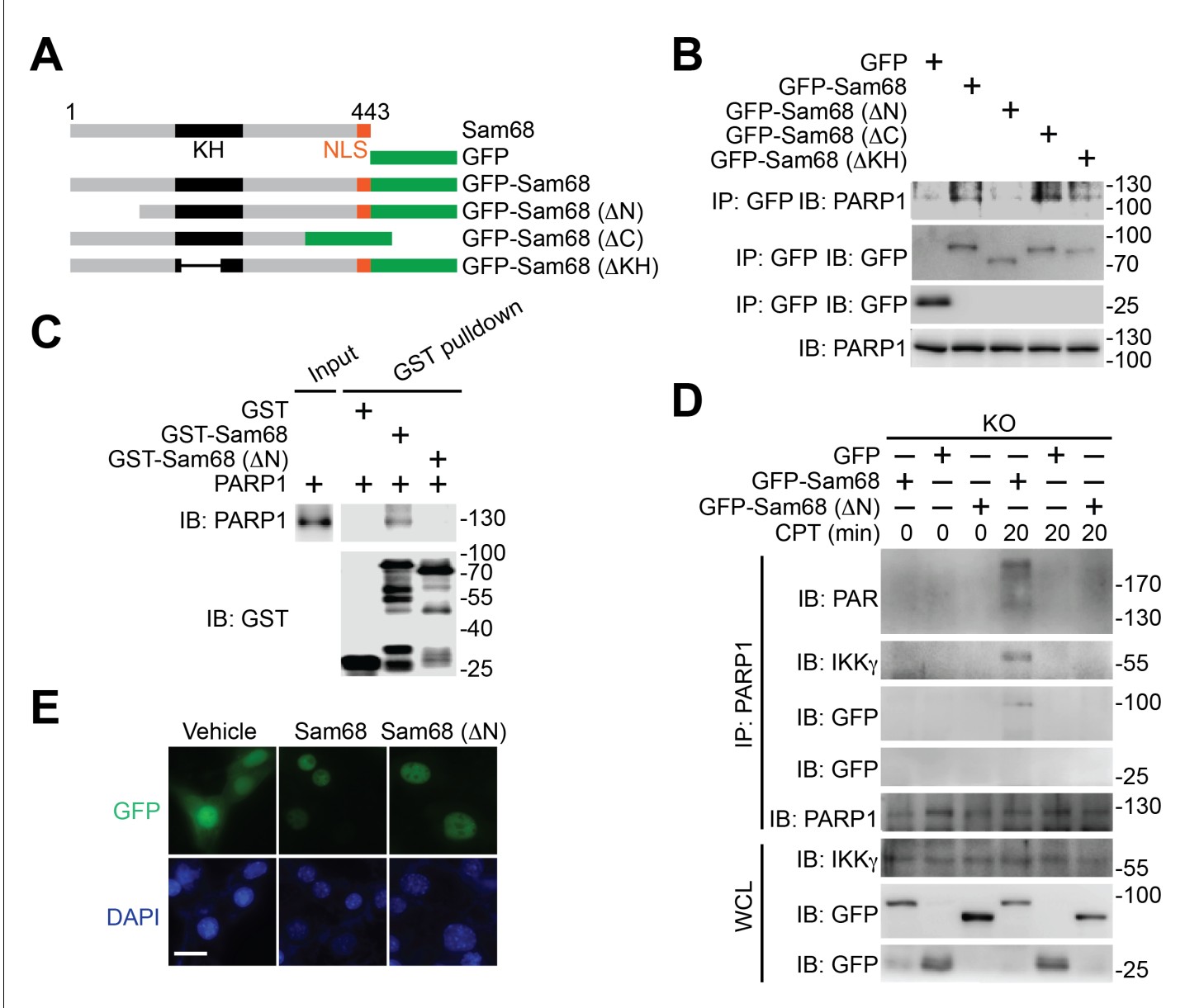

**Figure 3.** N-terminus of Sam68 is crucial for the Sam68-PARP1 interaction and genotoxic stress-induced NF-κB signalosome assembly. (**A**) Schematic diagram of Sam68 protein (residues 1–443), full-length or indicated mutants (ΔN lacks residues 1–102, ΔC lacks 347–443, and ΔKH lacks 165–224) fused with GFP. KH, The hnRNP K homology (KH) domain and nuclear localization signal (NLS) are labeled in black and orange, respectively. (**B**) Whole cell lysates from HEK293T cells expressing indicated GFP or GFP-fusion proteins were IB directly or after IP with GFP antibody for indicated proteins. (**C**) Whole cell lysate (Input) containing recombinant PARP1 were IB directly or after pulldown with indicated GST or GST-fusion proteins for indicated proteins. (**D**) Sam68 KO MEFs expressing GFP, GFP-Sam68, or GFP-Sam68 (ΔN) proteins were stimulated with 10 μM of CPT for indicated periods, and the derived whole cell lysates (WCL) were IB directly or after IP with PARP1 antibody for indicated proteins. (**E**) Immunofluorescence micrographs of Sam68 KO MEFs expressing GFP (Vehicle), GFP-Sam68 (Sam68), or GFP-Sam68 (ΔN) proteins, with nuclei counterstained by DAPI. Scale bar, 10 μm. Data in (**B–E**) are representative of at least three independent experiments.

The following figure supplement is available for figure 3:

**Figure supplement 1.** The N-terminus-mediated Sam68-PARP1 interaction is critical for DNA damage-induced PARylation and NF-κB activation.

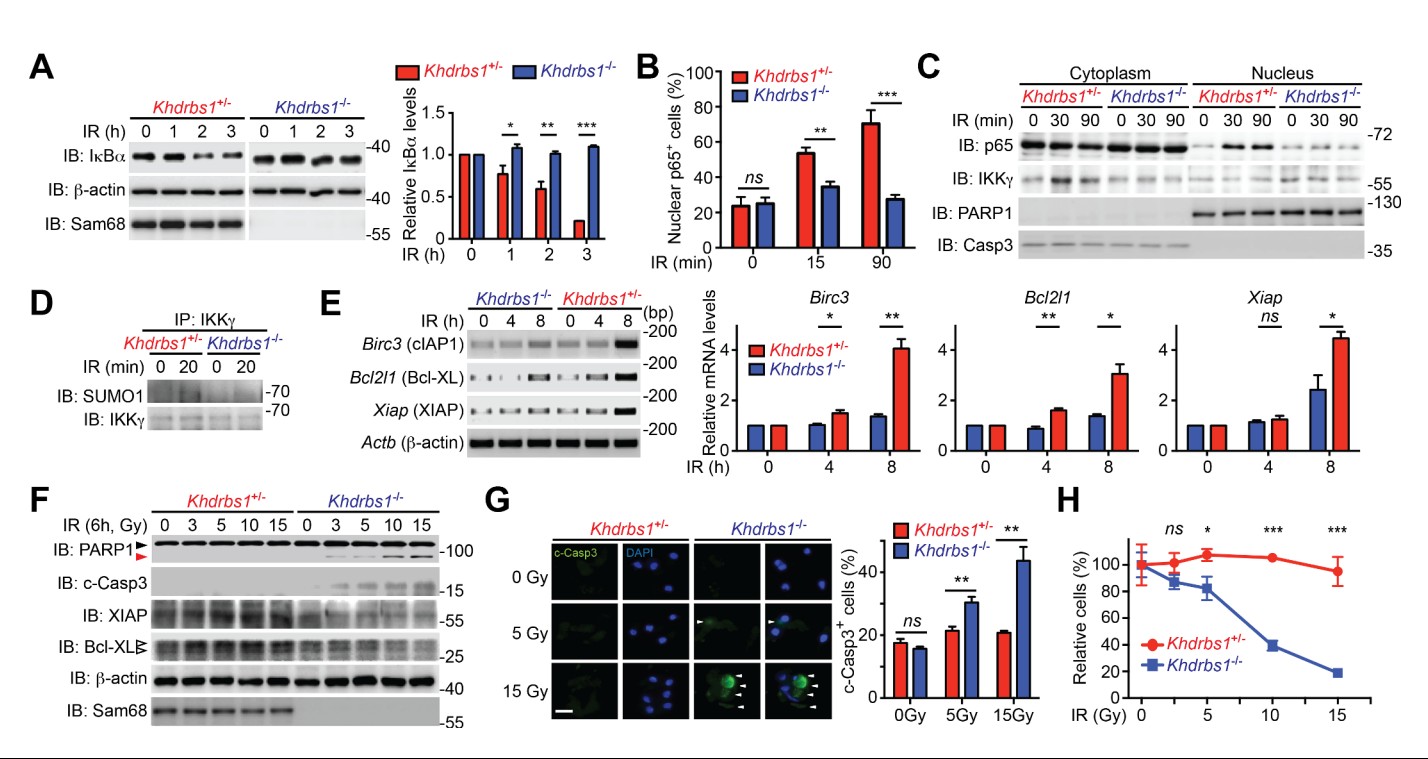

**Figure 4.** Sam68 deficiency attenuates NF-κB-mediated anti-apoptotic transcription in mouse colonic epithelial cells (CECs) under genotoxic stresses and sensitizes CECs to death. (A) Whole cell lysates from primary *Khdrbs1*[+/-] and *Khdrbs1*[-/-] CECs treated with 10 Gy of γ-irradiation (IR) for indicated periods were immunoblotted (IB) for IκBα and Sam68, with β-actin as a loading control. *Right*, the IκBα levels, normalized to β-actin and untreated controls, were quantified from three independent experiments. (B) Isolated *Khdrbs1*[+/-] and *Khdrbs1*[-/-] CECs were treated with 10 Gy of IR for indicated periods, and fixed cells were stained for p65 and nuclei and subjected to immunofluorescence micrographs. Percentage of CECs (>100 cells from 5–8 random fields) with nuclear p65 staining was quantified. (C) Cytosolic and nuclear fractions derived from *Khdrbs1*[+/-] and *Khdrbs1*[-/-] CECs treated as in (B) were IB for indicated proteins. Cytosolic Caspase-3 (Casp3) and PARP1 served as loading controls and cytosolic and nuclear markers, respectively. (D) Whole cell lysates from *Khdrbs1*[+/-] and *Khdrbs1*[-/-] CECs stimulated with 10 Gy of IR for indicated periods, were IB for indicated proteins after immunoprecipitation (IP) with IKKγ antibody. (E) Total RNA was extracted from *Khdrbs1*[+/-] and *Khdrbs1*[-/-] CECs at indicated time points following IR (10 Gy) and mRNA profiles of *Birc3, Bcl2l1, Xiap*, and *Actb* were analyzed by semi-quantitative RT-PCR. *Right*, the relative expression levels of *Birc3, Bcl2l1* and *Xiap*, normalized to *Actb* and untreated controls, were quantified from three independent experiments. (F) *Khdrbs1*[+/-] and *Khdrbs1*[-/-] CECs were γ-irradiated with indicated doses for 6 hr, and whole cell lysates were derived and IB for indicated proteins. c-Casp3, cleaved Caspase-3. The full-length and cleaved PARP1 are indicated by a black triangle and a red triangle, respectively; the two species of Bcl-XL proteins are labeled by open triangles. (G) Immunofluorescence micrographs of c-Casp3 in CECs treated as in (F), with nuclei counterstained by DAPI. Scale bar, 10 μm. Percentage of CECs (>100 cells from 5–8 random fields) with c-Casp3 staining was quantified (right). (H) *Khdrbs1*[+/-] and *Khdrbs1*[-/-] CECs were treated as in (F), and live cells following γ-irradiation were counted using a particle counter and normalized to the un-irradiated controls. Results in (A, B, E, G and H) are expressed as mean and s.e.m. ns, non-significant difference; *p<0.05; **p<0.01; ***p<0.001 by Student's t tests. Data are representative of at least three independent experiments.

The following figure supplements are available for figure 4:

**Figure supplement 1.** Sam68 deletion attenuates genotoxic stress-induced NF-κB signaling cascade in primary mouse cells.

**Figure supplement 2.** Sam68 deficiency attenuates DNA damage-triggered NF-κB-mediated expression of anti-apoptotic molecules.

cytoplasmic NF-κB liberation (*McCool and Miyamoto, 2012*; *Miyamoto, 2011*); however, such signaling events was dampened in γ-irradiated *Khdrbs1*[-/-] cells (*Figure 4C–D*). These results, together with our findings that Sam68 deletion diminishes DNA damage-initiated PARylation in CECs (*Figure 2B–D*), suggest that Sam68 is essential in the nuclear-initiated NF-κB signaling in mouse primary cells.

## Sam68 is pivotal for NF-κB-mediated anti-apoptotic transcription in mouse CECs

NF-κB-mediated transcription of a panel of anti-apoptotic molecules is an important factor for cell fate determination after DNA damage (*Chen et al., 2015*; *Kim et al., 2005*; *Stilmann et al., 2009*). Indeed, mRNA levels of *Birc3* (encoding cellular inhibitor of apoptosis protein-1, cIAP1), *Bcl2l1* (encoding B-cell lymphoma-like-1, Bcl-XL), and *Xiap* (encoding X-linked inhibitor of apoptosis protein, XIAP) were elevated in Sam68 sufficient CECs and MEFs post γ-irradiation (*Figure 4E* and *Figure 4—figure supplement 2A*). However, γ-irradiation-induced transcription of these genes was attenuated in Sam68 KO CECs and MEFs (*Figure 4E* and *Figure 4—figure supplement 2A*), in line with attenuated nuclear-initiated NF-κB signaling (*Figure 4A–D*). Moreover, XIAP and Bcl-XL protein levels were enhanced in *Khdrbs1*$^{+/-}$ CECs (*Figure 4F*) and wild-type MEFs (*Figure 4—figure supplement 2B*) following γ-irradiation, whereas such induction did not occur in *Khdrbs1*$^{-/-}$ CECs and MEFs (*Figure 4F* and *Figure 4—figure supplement 2B*). In contrast, cleavage of PARP1 and Caspase-3, two known biochemical hallmarks for apoptosis, were elevated in the γ-irradiated *Khdrbs1*$^{-/-}$, but not *Khdrbs1*$^{+/-}$ CECs (*Figure 4F–G*), mirroring the inefficient anti-apoptotic gene expression in *Khdrbs1*$^{-/-}$ CECs (*Figure 4E–F*). Consistently, more *Khdrbs1*$^{-/-}$ CECs underwent cell death in response to γ-irradiation than *Khdrbs1*$^{+/-}$ controls, as reflected by an irradiation dose-dependent cell loss (*Figure 4H*). Thus our data suggest that Sam68 deficiency diminishes nuclear-initiated NF-κB signaling, thus dampening NF-κB-mediated anti-apoptotic gene transcription and promoting cells to undergo cell death.

## Sam68 protein levels are elevated in colon tumors from *Apc*$^{min716/+}$ mice and human patients

It is widely accepted that massive intrinsic DNA damage occurs during rapid DNA replication and proliferation in cancer cells (*Hanahan and Weinberg, 2011*; *Rouleau et al., 2010*), and that the dysregulation of NF-κB and apoptosis play crucial roles in cancer development and progression (*Townson et al., 2003*; *Zubair and Frieri, 2013*). We therefore examined the relevance of Sam68 and DNA damage-initiated NF-κB signaling in *Apc*$^{min716/+}$ mice, a mouse model for human colon cancer (*Wu et al., 2009*). Colon adenomas spontaneously developed in *Apc*$^{min716/+}$ mice, as conveyed by staining with hematoxylin/eosin and the colon cancer marker β-catenin (*Figure 5A*). Interestingly, Sam68 levels were elevated in colon tumors, compared to adjacent normal tissue, from the tumor-laden *Apc*$^{min716/+}$ mice (*Figure 5B and D*). Moreover, the enhanced Sam68 expression coincided with elevated levels of PAR production, phosphorylated p65 (phosphor-p65, indicative of NF-κB activation), and Bcl-XL (anti-apoptotic transcriptional target of NF-κB) in colon tumors (*Figure 5B–D*). Similarly, in tissue samples derived from colon adenocarcinoma patients (*Figure 5E* and *Figure 5—source data 1*), Sam68 levels were substantially elevated in 16 (94.1%) out of 17 patients, when compared to adjacent normal tissue from the same patient (*Figure 5G–H*). Moreover, anti-apoptotic molecules Bcl-XL and XIAP were both upregulated at the transcriptional (5 [100%] out of 5 patients) and translational (16 [94.1%] out of 17 patients) levels in colon tumors, in comparison to normal tissue controls (*Figure 5F–H*). Furthermore, the elevated Sam68 levels positively correlated with increased PAR and phospho-p65 levels in human colon cancer samples (*Figure 5H–I*). Therefore these correlative results suggest that elevated Sam68 levels could facilitate PAR synthesis and PAR-dependent NF-κB signaling/transactivation of anti-apoptotic genes to counter intrinsic DNA damage in human and mouse colon tumor cells.

## Sam68 is essential for colon tumor development and survival

To assess the impact of Sam68 in colon tumor development and survival, we examined colon adenoma development in *Apc*$^{min716/+}$ mice in the presence and absence of Sam68. *Apc*$^{min716/+}$; *Khdrbs1*$^{+/-}$ mice, compared to wild-type controls, spontaneously developed adenomas in the cecum and distal colon at 3 months of age, as conveyed by whole mount methylene blue staining (*Figure 6A*) and showed substantial increases in the size and load of tumors (*Figure 6B*). In contrast, colon tumor development, as reflected by both tumor size and tumor load, in *Apc*$^{min716/+}$; *Khdrbs1*$^{-/-}$ mice was dramatically reduced; albeit genetic deletion of Sam68 in *Apc*$^{min716/+}$ mice did not significantly affect tumor number (*Figure 6B*). These results illustrate an essential role of Sam68 in the colon tumor growth and survival in *Apc*$^{min716/+}$ mice.

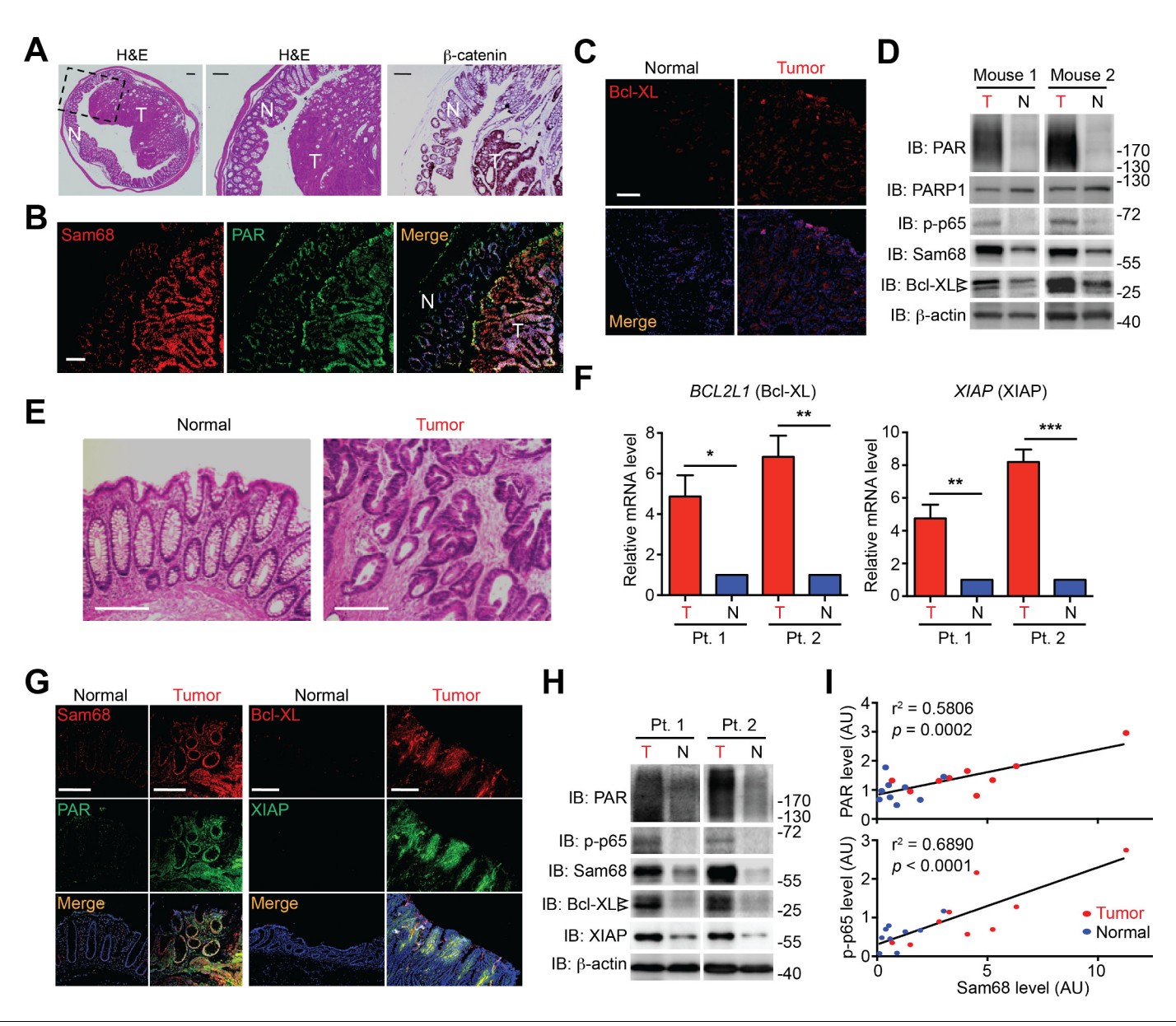

**Figure 5.** Sam68, PAR, and NF-κB-mediated anti-apoptotic transcription are elevated in mouse and human colon cancers. (A and E) Hematoxylin and eosin (H&E) staining and β-catenin immunohistochemistry of colon sections from tumor-loaded $Apc^{min716/+}$ mice (A) and tissue sections of colon tumor or adjacent normal colon tissue from human cancer patients (E). Scale bars, 200 μm. N, normal tissue; T, tumor tissue. (B, C and G) Immunofluorescence micrographs of indicated proteins in colon sections from tumor-loaded $Apc^{min716/+}$ mice (B, C) or Normal and Tumor tissue derived from human colon cancer patients (G), with nuclei counterstained by DAPI. Scale bars, 100 μm. (D and H) Colonic epithelial cells were isolated from normal (N) or tumor (T) colon tissue from tumor-loaded $Apc^{min716/+}$ mice (D) or normal (N) and tumor (T) tissue derived from human colon cancer patients (Pt.) (H) and whole cell lysates were derived and immunoblotted for indicated proteins, with β-actin as a loading control. The two species of Bcl-XL proteins are labeled by open triangles. (F) Relative mRNA levels of *BCL2L1* and *XIAP*, normalized to *ACTB*, from normal (N) and tumor (T) tissue derived from human colon cancer patients (Pt.). (I) Linear regression analysis of the levels of Sam68 protein versus PAR and phosphorylated p65 in CECs from normal (blue) and tumor (red) tissue derived from human colon cancer patients. AU, arbitrary unit. Results in (F) are expressed as mean and s.e.m. ns, non-significant difference; *p<0.05; **p<0.01; ***p<0.001 by Student's *t* tests. Data in (A–H) are representative of at least three independent experiments.

The following source data is available for figure 5:

**Source data 1.** Surgical colorectal cancer (CRC) and polyp metadata.

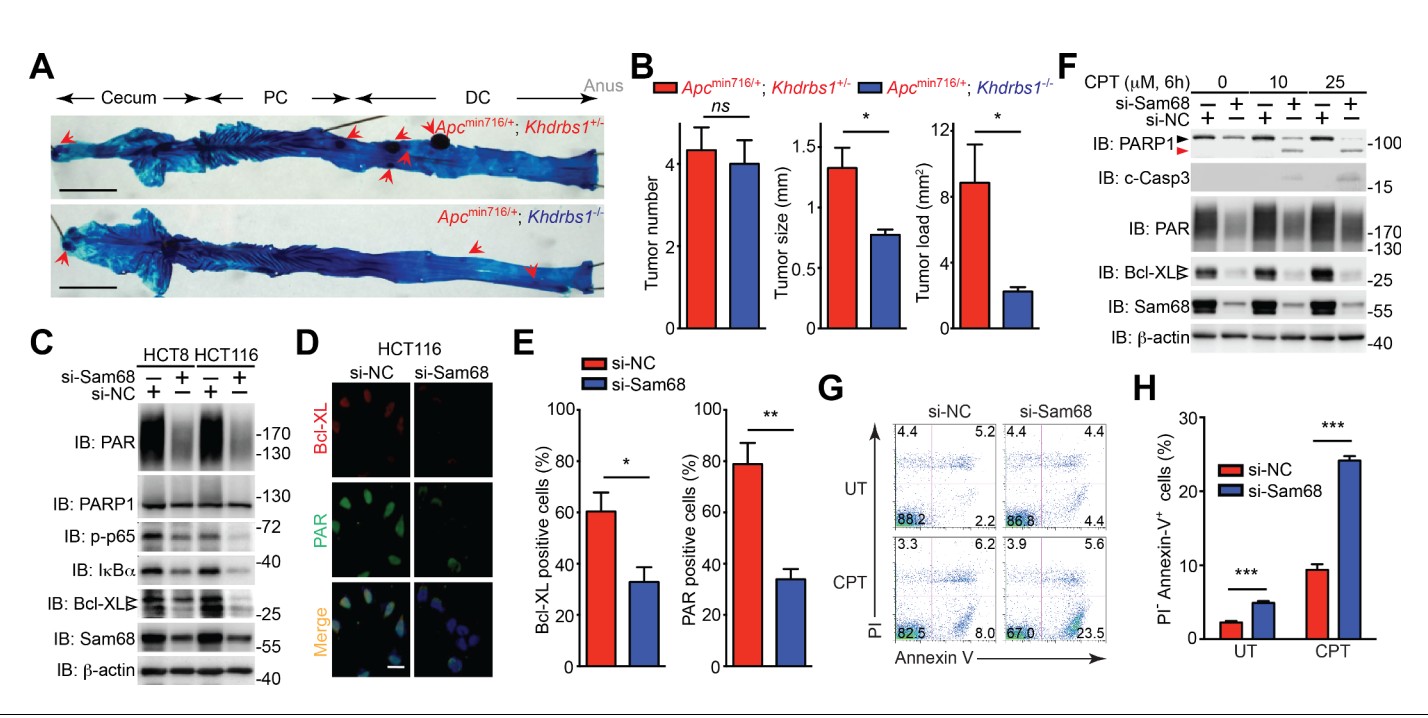

**Figure 6.** Sam68 plays a critical protective role for the survival of mouse and human colon cancers. (A) Methylene blue (MB) staining of the colons (with cecum, proximal colon [PC], distal colon [DC], and anus indicated) derived from 3-month old *Apc*^min716/+^; *Khdrbs1*^+/-^ and *Apc*^min716/+^; *Khdrbs1*^-/-^ mice. Red arrows indicate colon tumors. Scale bar, 1 cm. (B) Quantification of tumor number, tumor size, and tumor load in the colons from *Apc*^min716/+^; *Khdrbs1*^+/-^ (n = 6) and *Apc*^min716/+^; *Khdrbs1*^-/-^ mice (n = 3) following MB staining. (C) HCT8 and HCT116 cells were transfected with nonspecific control (si-NC) or Sam68-specific (si-Sam68) small interference RNAs. 72 hr later, whole cell lysates were derived and immunoblotted (IB) for indicated proteins, with β-actin as a loading control. (D) Immunofluorescence micrographs of Bcl-XL and PARylated (PAR) proteins in the si-NC and si-Sam68 transfected HCT116 cells, with nuclei counterstained by DAPI. Scale bar, 20 μm. (E) Percentage of HCT116 cells (>100 cells from 5–8 random fields) with Bcl-XL and PAR staining was quantified. (F) HCT116 cells transfected with indicated siRNAs as in (C) were stimulated with indicated doses of Camptothecin (CPT) for 6 hr. Whole cell lysates were derived and IB for indicated proteins, with β-actin as a loading control. c-Casp3, cleaved Caspase-3. The full-length and cleaved PARP1 are indicated by a black triangle and a red triangle, respectively; the two species of Bcl-XL proteins are labeled by open triangles. (G) HCT8 cells transfected with indicated siRNAs as in (C) were left untreated (UT) or stimulated with 10 μM of CPT for 12 hr, followed by propidium iodide (PI)/Annexin V staining and flow cytometry analysis. (H) Percentages of apoptotic (PI⁻ Annexin V⁺) HCT8 cells treated as in (G) were quantified. Results in (B, E, and H) are expressed as mean and s.e.m. ns, non-significant difference; *p<0.05; **p<0.01; ***p<0.001 by Student's *t* tests. Data in (A and C–H) are representative of at least three independent experiments.

The following figure supplements are available for figure 6:

**Figure supplement 1.** Sam68 knockdown and PARP inhibition attenuate PAR synthesis and PAR-dependent NF-κB transactivation in human colon cancer cell lines.

**Figure supplement 2.** Sam68 knockdown sensitizes human colon cancer cells to genotoxic stress-induced apoptosis.

We further examined whether Sam68 is essential for human colon cancer cell survival, knowing that Sam68 levels were elevated in colon cancer from human patients (*Figure 5G–I*). Indeed, knockdown of Sam68 by siRNAs significantly reduced PAR levels in human colon cancer-derived HCT8, HCT116, and T84 cell lines (*Figure 6C–E* and *Figure 6—figure supplement 1A–C*). Moreover, p65 phosphorylation and Bcl-XL expression were substantially attenuated in Sam68 knockdown cancer cells (*Figure 6C–E* and *Figure 6—figure supplement 1A–C*). Furthermore, genotoxic stress-dependent NF-κB activation, as indicated by p65 phosphorylation and ATM phosphorylation (*Wu et al., 2006*) (*Figure 6—figure supplement 1D*) and subsequent Bcl-XL expression (*Figure 6F*) were both significantly attenuated in Sam68 knockdown cancer cells. These results thus suggest a critical role of Sam68 in DNA damage-initiated and PAR-dependent NF-κB transactivation. Along with reduced anti-apoptotic transcription, Sam68 knockdown sensitized cancer cells to CPT- or γ-irradiation-

induced apoptosis, as conveyed by the boosted cleavage of PARP1 and Caspase-3 (*Figure 6F* and *Figure 6—figure supplement 2A–C*). Of note, ectopic expression of IKKβ (SSEE), which constitutively activates NF-κB, substantially rescued the attenuated Bcl-XL expression and the DNA damage-induced apoptosis, as evidenced by reduced PARP1 and Caspase-3 cleavage, in Sam68 downregulated HCT116 cells (*Figure 6—figure supplement 2D*), which supports that Sam68 knockdown affects genotoxic stress-induced NF-κB transactivation. Consistently, in contrast to controls, Sam68 knockdown triggered colon cancer cells to undergo spontaneous apoptosis and dramatically sensitized cancer cells to CPT- or γ-irradiation-induced cell death, as conveyed by Annexin-V staining (*Figure 6G–H*). These results thus demonstrate that downregulation of Sam68 lessens colon tumor development in *Apc*^min716/+ mice and sensitizes human colon cancer cells to genotoxic stress-induced apoptosis, in line with the indispensible role of Sam68 in the nuclear-initiated PARylation, NF-κB activation, and anti-apoptotic transcription in mouse and human colon cancer cells.

## PARP1 and NF-κB transactivation are critical for colon tumor development and survival

To ascertain whether Sam68 deletion reduces colon tumor formation as a result of the defect in PARP1 activation and PARylation, we assessed the impact of the PARP inhibitor, Olaparib, on phosphor-p65 and Bcl-XL levels and colon tumor development in *Apc*^min716/+ mice. 8-week-old *Apc*^min716/+ mice (when visible colon adenomas start to form) were utilized to assess whether inhibiting PAR production and PAR-dependent NF-κB signaling and anti-apoptotic transcription prevents adenoma formation. Of note, five continuous daily intraperitoneal injections of Olaparib, in comparison to vehicle control, substantially reduced PAR production and levels of phosphorylated p65 and Bcl-XL in CECs derived from *Apc*^min716/+ mice (*Figure 7A–B*), indicative of the impact of PARP1 inhibition on PAR formation and NF-κB transactivation in vivo. Consistent with our observation (*Figure 6A–B*), the vehicle-treated *Apc*^min716/+ mice spontaneously developed adenomas in the cecum and distal colon (*Figure 7C–D*). In contrast, a 4-week treatment with Olaparib significantly retarded colon tumor development in *Apc*^min716/+ mice (*Figure 7C–D*). PARP1 inhibition by Olaparib, compared to the control, substantially reduced the tumor size and tumor load, albeit had no statistically significant impact on tumor number was observed in the tumor-laden *Apc*^min716/+ mice (*Figure 7E*). Thus genetic deletion of Sam68 and PARP1 inhibition exhibited similar effects on reducing colon tumor development in *Apc*^min716/+ mice, which supports the essential roles of Sam68 and PARP1 in mouse tumor growth in vivo.

We sought to examine whether down-regulation of PARP1 or NF-κB transactivation executes a similar function as down-regulation of Sam68 on human colon cancer cell survival. Similar to Sam68 knockdown (*Figure 6C–E* and *Figure 6—figure supplement 1*), PARP1 inhibition by Olaparib or PJ-34 treatment significantly reduced the basal PAR, phospho-p65, and Bcl-XL levels in HCT116 cells (*Figure 7F* and *Figure 7—figure supplement 1A*) as well as genotoxic stress-induced Bcl-XL expression (*Figure 7—figure supplement 1B*). In line with the attenuated NF-κB activation and anti-apoptotic transcription, Olaparib or PJ-34 treatment triggered colon cancer cells to undergo spontaneous apoptosis, as conveyed by Annexin-V staining (*Figure 7G* and *Figure 7—figure supplement 1C–D*), which mirrors the impact of Sam68 knockdown on the survival of these cell lines (*Figure 6G–H*). Moreover, PARP1 knockdown by siRNAs, similar to Sam68 knockdown (*Figure 7—figure supplement 1E*), substantially reduced PAR production and levels of phosphorylated p65 and Bcl-XL in HCT116 cells (*Figure 7—figure supplement 1F*). In contrast, p65 knockdown attenuated only the levels of phosphorylated p65 and Bcl-XL, without affecting PAR production (*Figure 7—figure supplement 1G*), supporting the PAR-dependent NF-κB transactivation of anti-apoptotic genes. Importantly, down-regulation of PARP1 or p65 sensitized colon cancer cells to undergo spontaneous apoptosis (*Figure 7—figure supplement 1F–G*), which mirrors the effects of PARP1 inhibition (*Figure 7G* and *Figure 7—figure supplement 1C–D*) and Sam68 knockdown (*Figure 6G–H* and *Figure 7—figure supplement 1E*). Therefore, our results demonstrate the Sam68-PARP1-NF-κB-anti-apoptotic gene axis plays a crucial function for colon cancer survival.

## Discussion

The NF-κB signaling pathway remains a very attractive avenue for pharmacological intervention, given its crucial function in human health and disease, particularly inflammatory diseases and

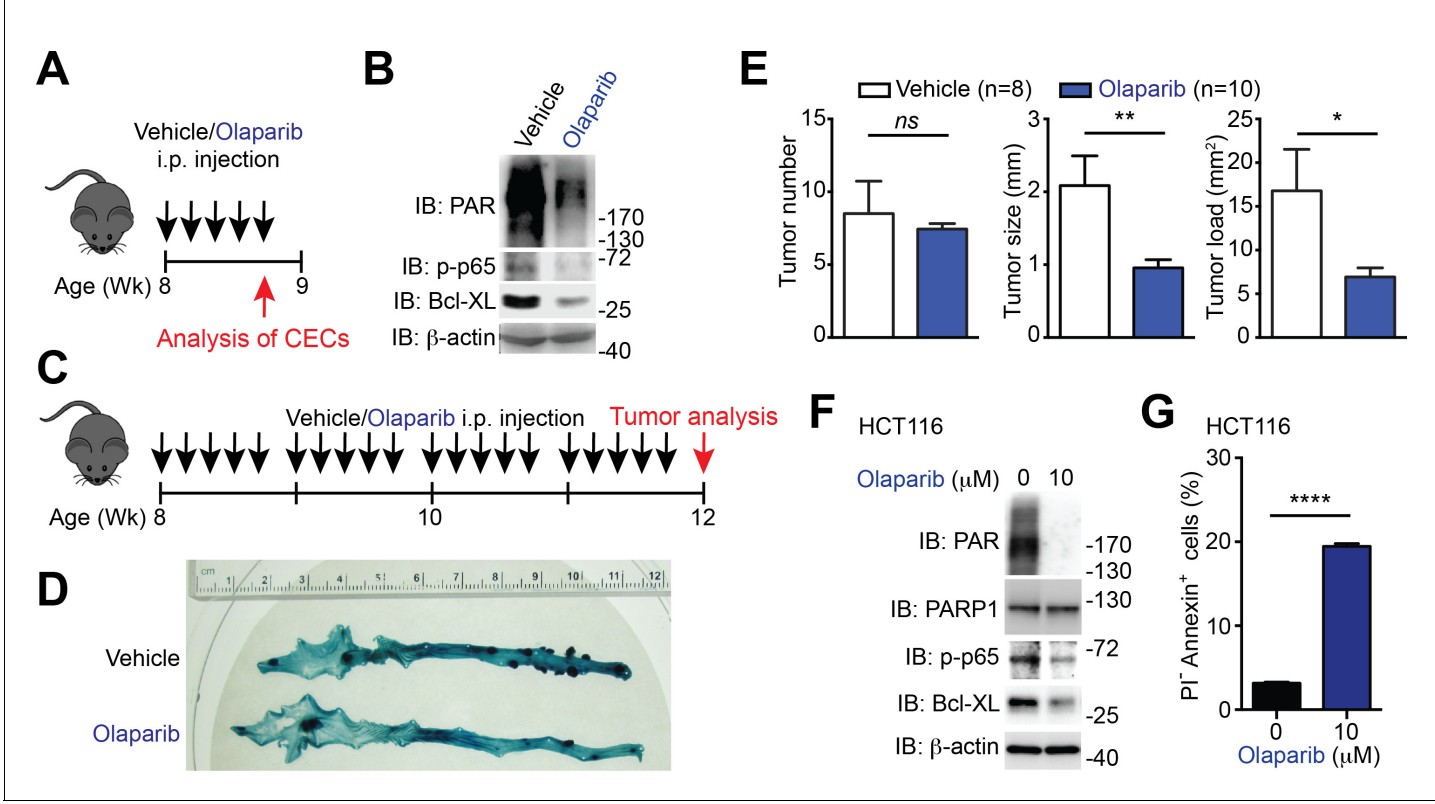

**Figure 7.** PARP1 inhibition reduces colon tumor development in mice and sensitizes human colon cancer cells to undergo apoptosis. (A) PARP1 inhibition in *Apc*^min716/+ mice in vivo. 8-week-old *Apc*^min716/+ mice were intraperitoneally injected with vehicle control or Olaparib (50 mg/kg) once daily for 5 days, followed by euthanization and further analysis. (B) Colon epithelial cells (CECs) were isolated from vehicle- or Olaparib-treated *Apc*^min716/+ mice, treated as in (A), and whole cell lysates were derived and immunoblotted (IB) for indicated proteins, with β-actin as a loading control. (C) A schematic of the experimental timeline for the impact of PARP1 inhibition on colon tumor development in vivo in *Apc*^min716/+ mice. 8-week-old *Apc*^min716/+ mice were intraperitoneally injected vehicle control or Olaparib (50 mg/kg, once daily for 5 days × 4 weeks). Mice were euthanized to analyze tumor development in the colon. (D) Methylene blue (MB) staining of the colons derived from 12-week old *Apc*^min716/+ mice, post 4-week vehicle control or Olaparib treatment, as illustrated in (C). (E) Quantification of tumor number, tumor size, and tumor load in the colons from *Apc*^min716/+ mice treated with vehicle control (n = 8) and Olaparib (n = 10), following MB staining. (F) HCT116 cells were treated with indicated concentration of Olaparib for 72 hr. Whole cell lysates were derived and IB for indicated proteins, with β-actin as a loading control. (G) HCT116 cells were treated with Olaparib as in (F), and subjected to flow cytometry analysis of propidium iodide (PI)/Annexin V staining. Percentages of apoptotic (PI⁻ Annexin V⁺) cells as indicated were quantified. Results in (E and G) are expressed as mean and s.e.m. ns, non-significant difference; *p<0.05; **p<0.01; ****p<0.0001 by Student's *t* tests. Data in (B, D, F, and G) are representative of at least three independent experiments.

The following figure supplement is available for figure 7:

**Figure supplement 1.** PARP1 inhibition and down-regulation of PARP1 and NF-κB triggers human cancer cells undergo apoptosis.

cancers. In spite of the widespread use of chemotherapy and radiotherapy in current-day cancer treatments, the genotoxic stress-induced nuclear NF-κB signaling pathway that leads to NF-κB trans-activation is still less defined, than NF-κB activation initiated from cell membrane stimuli (*e.g.* immune receptors). Herein, we report that Sam68 is a novel regulator participating in the early cellular responses to DNA damage; it does this by orchestrating the signaling cascade that links DNA lesion recognition in the nucleus to NF-κB liberation and activation in the cytoplasm (*Figure 8*). Following genotoxic stress, sophisticated cellular networks consisting of a variety of molecules and post-translational modifications, collectively termed as DNA damage responses (DDR), are crucial for cell-cycle checkpoint control, DNA repair, transcription, and apoptosis (*Jackson and Bartek, 2009*). Among these, DNA damage-initiated NF-κB signaling and transactivation of an array of anti-apoptotic molecules are pivotal in facilitating cells to 'escape' from the lethal effects of DNA damage (*McCool and Miyamoto, 2012*; *Miyamoto, 2011*). Our results demonstrate that Sam68 deletion

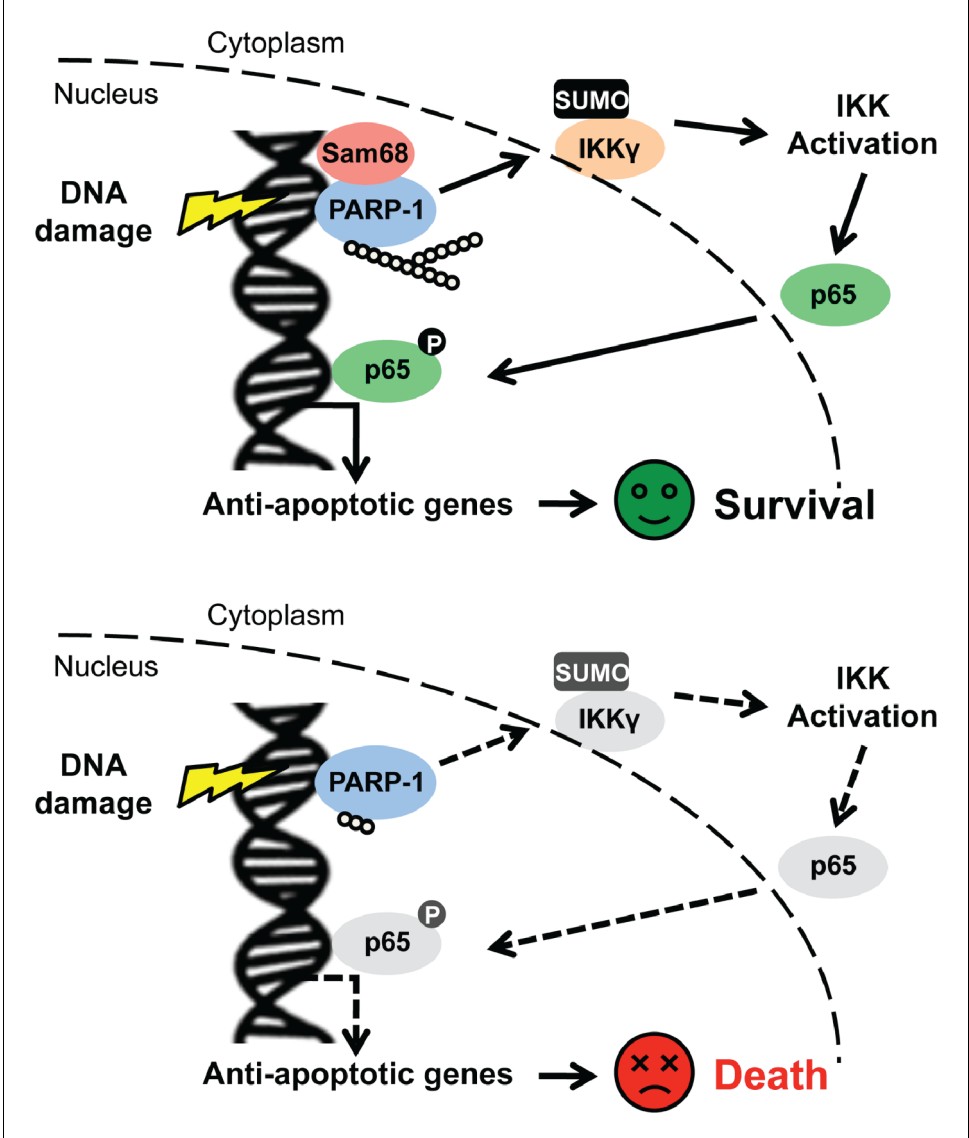

**Figure 8.** Schematic model representation of Sam68 functioning as an early signaling molecule in genotoxic stress-initiated NF-κB signaling pathway.

impairs inducible IκBα degradation, NF-κB liberation, and subsequent anti-apoptotic gene expression in cells under genotoxic stress. Sam68 interacts with several established molecules including PARP1 and IKKγ in the nuclear-initiated NF-κB signaling pathway and facilitates assembly of the NF-κB activation signaling complex. More importantly, Sam68 plays an indispensable role in DNA damage-triggered PARP1 activation and PAR synthesis, thus controlling the PAR-dependent signaling complex assembly. This is distinct from leukemia related protein 16 (LRP16), a recently reported regulator of DNA damage-induced NF-κB activation, which binds to PARP1 and IKKγ in a PAR-dependent manner and therefore functions downstream of PARP1 and PAR production (*Wu et al., 2015*). Previously, PARP1 was proposed to be dispensable for ATM activation (*Stilmann et al., 2009*). Our results suggest that Sam68 is important for the activation of PARP1, as well as, ATM in DDR. In addition to the currently illustrated major impact of Sam68 on the PARP1/PAR-dependent signaling pathway that leads to NF-κB activation, Sam68 may contribute to NF-κB activation via a direct or indirect effect on the ATM-involved signaling cascade during the cellular response to DNA damage. Our results underscore a previously unknown function of Sam68, a versatile protein that preferentially

resides in the nucleus, in the early nuclear signaling cascade following DNA damage. Furthermore, we previously reported that Sam68 is important for the promoter selectivity and transcriptional specificity of NF-κB in the nucleus (*Fu et al., 2013*). Sam68 could work at both initiating of NF-κB activation and controlling the NF-κB transactivation potential in the DNA damage-induced NF-κB signaling pathway; thus playing a critical role in the genotoxic stress-initiated "nuclear to cytoplasmic to nuclear" NF-κB activation. Of note, tumor necrosis factor/tumor necrosis factor receptor (TNF/TNFR) signaling was revealed to be important for a feed-forward response to DNA damage, supported by DNA damage-induced phosphorylation of several well-known components of TNF/TNFR signaling pathway including TRAF2, p62, RIP1, and CYLD (*Beli et al., 2012*). Moreover, Sam68 has been proposed to be crucial for the recruitment of RIP1 to the TNF receptor in TNF-triggered NF-κB signaling (*Ramakrishnan and Baltimore, 2011*). The involvement of Sam68 in both TNF/TNFR signaling and genotoxic stress-induced NF-κB signaling indicates that it could be a crucial molecule conferring the crosstalk between TNF/TNFR signaling pathway and DNA damage responses.

We demonstrated that, compared to $Khdrbs1^{+/-}$ cells, $Khdrbs1^{-/-}$ CECs are hypersensitive to genotoxic stress, since Sam68 deletion abolishes the DNA damage-initiated NF-κB signaling and attenuates the inducible expression of anti-apoptotic genes. These results, along with the previous report that NF-κB-mediated anti-apoptotic transcription executes a protective function to CECs following γ-irradiation (*Egan et al., 2004*), highlight the pathophysiological relevance of the nuclear-initiated NF-κB signaling and transactivation in colonic cell survival from environmental acute DNA damage. Of note, nuclear-initiated NF-κB signaling plays a key role in cellular responses to intrinsic DNA damage, especially damage that occurs during rapid DNA replication and proliferation in cancer cells (*Hayakawa et al., 2009*; *Horst et al., 2009*; *Kim et al., 2010*; *Koh et al., 2011*; *Kojima et al., 2004*; *Onizawa et al., 2009*; *Pouyet et al., 2010*; *Shaked et al., 2012*; *Stark et al., 2007*; *Wan et al., 2011*; *Williams et al., 2008*). Aberrant NF-κB activation and elevated expression of NF-κB target genes, in particular those encoding anti-apoptotic molecules, have been acknowledged as key factors facilitating colon cancer survival and development (*Song et al., 2014*; *Townson et al., 2003*; *Zubair and Frieri, 2013*). Here we report that the levels of Sam68, phospho-p65 (indicative of NF-κB activation), anti-apoptotic molecules Bcl-XL and XIAP are all elevated in colon tumors in comparison to adjacent normal tissue derived from genetically susceptible $Apc^{min716/+}$ mice and human colon cancer patients. Of note, the positive correlation between Sam68 levels and PAR levels, as well as phospho-p65 levels in human colon cancers suggests that Sam68 could be essential for the development and survival of colon cancer, considering the pivotal function of Sam68 in orchestrating the intrinsic DNA damage-initiated NF-κB signaling and transactivation. In support of this notion, knockdown of Sam68, PARP1, and p65 in human colon cancer cells significantly reduces the basal and genotoxic stress-induced PAR production, NF-κB activation, and expression of anti-apoptotic molecules Bcl-XL and XIAP, thus leading to spontaneous and DNA damage-induced apoptosis in Sam68-downregulated colon cancer cells. Moreover, genetic deletion of Sam68 and inhibition of PARP1 markedly reduces the development and survival of colon tumors in $Apc^{min716/+}$ mice, further supporting the pivotal role of Sam68-conferred PAR-dependent NF-κB activation in colon tumorigenesis.

In spite of the crucial role of PARP1 in DNA damage-induced NF-κB activation, DNA repair, and other cellular responses, the precise mechanisms of the activation and regulation of PARP1 remains elusive. We show here that Sam68 deficiency significantly attenuates DNA damage-induced PARP1 activation and PAR production, which suggests that Sam68, as an early signaling regulator, governs the genotoxic stress-stimulated PARP1 activity. In particular, the PAR-dependent NF-κB signaling cascade is dampened in Sam68 deleted cells, as well as PARP1 knockout cells (*Stilmann et al., 2009*). Moreover, the reduction in anti-apoptotic gene expression and increase in genotoxic stress-induced apoptosis are observed in Sam68 knockout cells and PARP1 knockout cells (*Stilmann et al., 2009*), in line with the impeded PAR-dependent NF-κB signaling in these cells. Furthermore, Sam68 knockout and PARP1 inhibition both attenuates colon tumor development in $Apc^{min716/+}$ mice. Such similarity in the phenotypes of Sam68- and PARP1-deficient cells and animals in response to genotoxic stresses further supports the notion that Sam68 is a crucial regulator of PARP1 in cellular response to genotoxic stress. Elevated Sam68 levels correlate with tumor progression and poor prognosis in multiple cancer patients and overexpression of Sam68 has been proposed as a prognostic marker (*Chen et al., 2012*; *Liao et al., 2013*; *Song et al., 2010*; *Zhang et al., 2009*); however, the significance of Sam68 in tumorigenesis is still obscure. Here we report that Sam68 knockdown

markedly sensitizes colon cancer cells to genotoxic stress-induced cell death and Sam68 knockout substantially retards colon tumor burden and survival in $Apc^{min716/+}$ mice, which highlights the pivotal function of Sam68 in tumor development and survival. Importantly, we establish proof-of-concept showing that manipulation of Sam68 sensitizes colon cancer to DNA damage-induced apoptosis. As a key early signaling regulator at the proxy of the nuclear-initiated NF-κB signaling pathway, Sam68 could provide a novel target for therapeutics for cancers and other human diseases associated with impaired DNA damage responses.

## Materials and methods

### Ethics statement
The human patient study was approved by the Johns Hopkins Institutional Review Board. All samples were obtained in accordance with the Health Insurance Portability and Accountability Act (HIPAA). All animal experiments were performed according to protocol number MO13-H349, approved by the Johns Hopkins University's Animal Care and Use Committee and in direct accordance with the NIH guidelines for housing and care of laboratory animals.

### Patient selection and sample acquisition
Colon tumors (adenomas and cancers) and paired normal tissues were collected from patients undergoing surgery at Johns Hopkins Hospital, as described previously (*Dejea et al., 2014*). Patients who received pre-operative radiation and/or chemotherapy or with a personal history of colitis-associated colon cancer were excluded.

### Mice
*Khdrbs1*⁻/⁻ (Sam68 knockout) mice and their gender-matched littermate *Khdrbs1*⁺/⁻ (Sam68 heterozygote) mice (occasionally substituted with gender-matched littermate *Khdrbs1*⁺/⁺ [Sam68 wild-type] mice when *Khdrbs1*⁺/⁻ ones were lacking, and referred as *Khdrbs1*⁺/⁻ alone for simplicity) were produced using heterozygous breeding pairs, as previously described (*Huot et al., 2012*). $Apc^{min716/+}$ mice expressing a mutant gene encoding an adenomatous polyposis coli protein truncated at amino acid 716 were described previously (*Su et al., 1992*; *Wu et al., 2009*). Mice were maintained in a specific pathogen-free facility and fed autoclaved food and water *ad libitum*.

### Cell culture, antibodies, and reagents
The following mouse embryonic fibroblasts (MEFs) were obtained from other institutions: wild-type and Sam68 knockout (KO) MEFs (*Richard et al., 2005*) from Stephan Richard (McGill University, Canada) and PARP1 KO MEFs (*Tong et al., 2001*) from Zhao-Qi Wang (Fritz Lipmann Institute, Germany), respectively. HEK293T, HCT8, HCT116, and T84 cell lines were purchased from ATCC (Manassas, VA) and the identities have been authenticated by short tandem repeat DNA profiling. All cells described above were regularly tested for mycoplasma contamination. Cells were cultured in DMEM medium containing 10% fetal calf serum, 2 M glutamine, and 100 U/ml each of penicillin and streptomycin. Antibodies used were: IκBα, Sam68, p65, IKKγ, GFP, PARP1, PARP2, GST from Santa Cruz Biotechnology (Dallas, TX); β-actin from Sigma-Aldrich (St. Louis, MO); PAR from Trevigen (Gaithersburg, MD); ATM, phospho-ATM, PARP1, phospho-p65, cleaved Caspase-3, Caspase-3, Ku70, Histone H3 from Cell Signaling Technology (Danvers, MA); XIAP form BD Biosciences (San Jose, CA); α-tubulin from EMD Millipore (Billerica, CA); Bcl-XL, cIAP1, NBS1 from GeneTex (Irvine, CA); SUMO1, kindly provided by Dr. M. Matunis (Johns Hopkins University). 4-[(3-[(4-cyclopropylcarbonyl)piperazin-4-yl]carbonyl)-4-fluorophenyl]methyl(2H)phthalazin-1-one (Olaparib) and *N*-(6-oxo-5,6-dihydrophenanthridin-2-yl)-*N*, *N*-dimethylacetamide-HCl (PJ-34) were purchased from Fisher Scientific (Pittsburgh, PA) and Enzo Life Sciences (Farmingdale, NY), respectively. Camptothecin (CPT), MG132, and 4',6-diamidino-2-phenylindole (DAPI) were obtained from Sigma-Aldrich. Recombinant PARP1 protein was obtained from Trevigen. The FLAG, FLAG-IKKβ (SSEE), GFP, GFP-Sam68, GFP-Sam68 (ΔC), GFP-Sam68 (ΔN), GFP-Sam68 (ΔKH), GST, GST-Sam68, and GST-Sam68 (ΔN) constructs were described previously (*Fu et al., 2013*).

## RNA interference and transfection

Mouse Sam68 siGENOME SMARTpool siRNA (catalog number M-065115-01) was purchased from Thermo Scientific (Waltham, MA). Human Sam68 and p65 siRNAs were described previously (*Fu et al., 2013*). Human PARP1 and PARG siRNAs were purchased from Santa Cruz Biotechnology. Transient transfection of siRNA or plasmids into MEFs was performed using Lipofectamine 2000 or Lipofectamine RNAiMAX (Life Technologies, Grand Island, NY) according to the manufacturer's instructions.

## Subcellular fractionation and electrophoretic mobility shift assays (EMSAs)

Subcellular fractionation was performed by differential centrifugation as previously described (*Wan et al., 2007*). EMSAs were carried out as described (*Wan et al., 2007*), and the reaction mixture was resolved on 6% DNA retardation gel (Life Technologies) in 0.25 × TBE buffer, and dried gels were visualized in a Fujifilm image reader FLA-7000 (Fujifilm Life Science, Valhalla, NY).

## Immunoprecipitation and immunoblot

Immunoprecipitation and immunoblot assays were conducted as previously described (*Hodgson et al., 2015*). In brief, cells were harvested and lysed on ice by 0.4 ml of lysis buffer (50 mM Tris-HCl [pH 8.0], 150 mM NaCl, 1% NP-40 and 0.5% sodium deoxycholate, 1 × complete protease inhibitor cocktail [Roche Applied Science, Indianapolis, IN]) for 30 min. The lysates were centrifuged at 10,000 ×*g* at 4°C for 10 min. The protein-normalized lysates were subjected to immunoprecipitation by adding 10 mg/ml of the appropriate antibody, 30 μl of protein G-agarose (Roche Applied Science), and rotating for more than 2 hr in the cold room. The precipitates were washed at least four times with cold lysis buffer followed by a separation by SDS-PAGE under reduced and denaturing conditions. The resolved protein bands were transferred onto nitrocellulose membranes and probed by the Super Signaling system (Thermo Scientific) according to the manufacturer's instructions, and imaged using a FluorChem E System (Protein Simple, Santa Clara, CA).

## Immunofluorescence microscopy

Immunofluorescence microscopy was performed as previously described (*Hodgson et al., 2015*). Briefly, cells were fixed with 4% paraformaldehyde in PBS and then mounted onto slides by Cellspin. After a 5-min permeabilization with 0.05% Triton X-100 in PBS, the fixed cells were stained with appropriate primary antibodies for 1 hr, and FITC- or AlexaFluor 594-conjugated secondary antibodies (Life Technologies) for 1 hr together with 1 μg/ml of DAPI for 5 min at 25°C. The slides were then rinsed with PBS three times and cover mounted for fluorescence microscopy.

## Chromatin fractionation

Cells were harvested at indicated time points after γ-irradiation, and cell pellets were resuspended in the NETN buffer (20 mM Tris–HCl [pH 8.0], 100 mM NaCl, 1 mM EDTA, and 0.5% NP-40) and incubated on ice for 20 min. Supernatant after 3000 ×*g* for 10 min were collected as soluble fraction. Pellets were recovered and resuspended in 0.2 M HCl on ice for 30 min, and sonicated for 10 sec to release chromatin-bound proteins, and then the soluble fractions were neutralized with 1 M Tris–HCl (pH 8.5) and collected as chromatin fraction, and the pellets were collected as insoluble fraction for further analysis, as described previously (*Liu et al., 2013*; *Wu et al., 2011*).

## Semi-quantitative reverse transcription PCR

Total RNA was isolated using Trizol reagent (Life Technologies) and treated with the TURBO DNA-free Kit (Life Technologies) to remove residual genomic DNA. Complementary DNA was synthesized using qScript cDNA SuperMix Kit (Quanta Biosciences, Gaithersburg, MD) according to the manufacturer's instructions. Gene specific products were amplified using MyTaq Rad Mix (Bioline USA, Taunton, MA) in a multiple conventional and gradient Veriti Thermal Cycler (Life Technologies) with the following primers: *Birc3*-f, 5′-GAAACCATTTGGCGTGTTCT-3′; and *Birc3*-r, 5′-TGGATCGCAATGA TGATGTC -3′; *Bcl2l1*-f, 5′-AATGAACTCTTTCGGGATGGAG-3′; and *Bcl2l1*-r, 5′- CCAACTTGCAA TCCGACTCA-3′; *Xiap*-f, 5′-CCATGTGTAGTGAAGAAGCCAGAT-3′; and *Xiap*-r, 5′-TGATCA

TCAGCCCCTGTGTAGTAG -3'; *Actb*-f, 5'-CACATCAAGAAGGTGGTG-3'; and *Actb*-r, 5'-TGTCA TACCAGGAAATGA-3'.

## In vitro PARylation assays

In vitro PARylation assays using recombinant His-PARP1 or immunoprecipitated endogenous PARP1 from MEFs were performed as previously described (*Zaniolo et al., 2007*). Briefly, PARP1 protein or immunoprecipitant was incubated for 20 min or 2 min at 30°C with GST or GST-Sam68 in a standard assay mixture containing 100 mM Tris-HCl (pH 8.0), 10 mM $MgCl_2$, 10% (v/v) glycerol, 1.5 mM DTT, 10 μg/ml activated DNA (sonicated) and 200 μM $NAD^+$. The reaction was terminated by the addition of SDS sample buffer (Life Technologies), and the boiled samples were subjected to SDS-PAGE. When indicated, the PARP inhibitor PJ-34 was added to the reaction mixture at a final concentration of 1 μM for 15 min prior to the reaction.

## Isolation of primary colonic epithelial cells

Colonic epithelial cells (CECs) were isolated from mice as previously described (*Hodgson et al., 2015*). Briefly, after euthanizing mice, the entire colon was removed under aseptic conditions and washed twice with ice-cold PBS. After dividing the colon into 2–3 mm long fragments and transferring them into chelating buffer (27 mM trisodium cirtcrate, 5 mM $Na_2HPO_4$, 96 mM NaCl, 8 mM $KH_2PO_4$, 1.5 mM KCl, 0.5 mM DTT, 55 mM D-sorbitol, 44 mM sucrose, 6 mM EDTA, 5 mM EGTA [pH 7.3]) for 15 min at 4°C, CECs were then dislodged by repeated vigorous shaking. Tissue debris was removed by a 70-μm cell-strainer (Fisher Scientific, Suwanee, GA) and CECs were harvested by centrifugation at 4°C. The viability of CECs was confirmed by trypan blue staining and isolated CECs were cultured at 37°C for 1 hr for recovery, followed by indicated treatment.

## γ-irradiation

The γ-irradiation on primary mouse cells and cell lines were performed using a [137]Caesium source (dose rate 8 Gy/min).

## Histology and immunohistology

After euthanizing mice, the colons were removed under aseptic conditions, washed once with ice-cold PBS, the terminal 0.5-cm piece of the colon was fixed in 10% buffered formalin for 24 hr, embedded in paraffin and 5-micron sections were cut and processed for Hematoxylin and Eosin (H&E) staining. For immunohistology, after euthanizing mice, the entire colons were excised under aseptic conditions and frozen in optimal cutting temperature (O.C.T.) media (Tissue-Tek, Elkhart, In). 5-micron frozen sections were cut using a Microm HM 550 Cryostat (Thermo Scientific), collected on coated slides, fixed in paraformaldehyde, washed with PBS, and blocked with appropriate sera in PBS. After incubating with appropriate antibodies, sections were washed and incubated with fluorescence dye-conjugated second antibodies and 1 μg/ml of DAPI. Stained sections were washed and mounted under a coverslip using Fluoro-gel with Tris Buffer (Electron Microscopy Sciences, Hatfield, PA) and examined using an Axio Observer fluorescence microscope (Zeiss, Oberkochen, Germany).

## Quantification of colon adenomas in mice

The visualization and quantification of colon adenomas in mice were conducted as previously described (*Wu et al., 2009*). Briefly, mice were sacrificed at 3 months of age. Colon tissue was excised, cleaned with cold PBS, opened longitudinally, fixed in 10% neutral buffered formalin (3.7% formaldehyde, 1.2% methanol, 6.5 g/l sodium phosphate dibasic, 4.0 g/l sodium phosphate monobasic) at 25°C overnight, and stained with 0.2% (w/v) methylene blue solution. The adenomas were quantified and sized under dissecting scope. Average tumor size and tumor load per individual mouse were determined by averaging diameters of all tumors present and summing the areas of all tumors presented in a given mouse, as previously described (*Grivennikov et al., 2012*).

## Statistical analyses

All statistical analysis was performed using GraphPad Prism version 6.0 (GraphPad Software, La Jolla, CA). Standard errors of means (s.e.m.) were plotted in graphs. Significant differences were

considered: ns, non-significant difference; * at p<0.05; ** at p<0.01; *** at p<0.001; **** at p<0.0001 by unpaired Student's *t*-test.

## Acknowledgements

We thank Drs. Johnny He, Anthony Leung, Michael Matunis, Stephane Richard, Bert Vogelstein, Zhao-Qi Wang, and Jihe Zhao for kindly sharing reagents and materials; Cory Brayton and Xin Guo for help with histological analyses; J Marie Hardwick for critical reading of the manuscript.

## Additional information

### Funding

| Funder | Grant reference number | Author |
| --- | --- | --- |
| National Cancer Institute | T32CA009110 | Eric M Wier |
| National Cancer Institute | R01CA151393 | Cynthia L Sears |
| National Institute of General Medical Sciences | R01GM111682 | Fengyi Wan |
| American Cancer Society | RSG-13-052-01-MPC | Fengyi Wan |

The funders had no role in study design, data collection and interpretation, or the decision to submit the work for publication.

### Author contributions

KF, XS, Designed and conducted most experiments, Acquisition of data, Analysis and interpretation of data; EMW, AH, YL, Helped with some experiments, Acquisition of data, Analysis and interpretation of data; CLS, Contributed to human patient study, Analysis and interpretation of data, Contributed unpublished essential data or reagents; FW, Conceptualized the study, Designed the experiments, Wrote the manuscript with input from all authors, Analysis and interpretation of data

### Author ORCIDs

Xin Sun, http://orcid.org/0000-0003-2424-8011
Fengyi Wan, http://orcid.org/0000-0001-9216-9767

### Ethics

Human subjects: The human patient study was approved by the Johns Hopkins Institutional Review Board. All samples were obtained in accordance with the Health Insurance Portability and Accountability Act (HIPAA).

Animal experimentation: All animal experiments were performed according to protocol number MO13-H349, approved by the Johns Hopkins University's Animal Care and Use Committee and in direct accordance with the NIH guidelines for housing and care of laboratory animals.

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
