## [Decision Letter]

Thank you for submitting your article "Sam68 is critical for colon tumorigenesis via regulating genotoxic stress-induced NF-κB activation" for consideration by *eLife*. Your article has been reviewed by three peer reviewers, and the evaluation has been overseen by a Reviewing Editor and Tony Hunter as the Senior Editor.

The reviewers have discussed the reviews with one another and the Reviewing Editor has drafted this decision to help you prepare a revised submission.

Summary:

Sam68 is an RNA-binding protein primarily localized in the nucleus that under particular circumstances can shuttle to the cytoplasm. Sam68 has been implicated in several physiological and pathological processes, from HIV replication to tumor progression. Although Sam68 has previously been shown to regulate tumor cell viability and carcinogenesis, the molecular mechanisms by which Sam68 regulates these process are poorly understood. Moreover, Sam68 regulates NF-κB signaling, a critical pathway in the cellular response to DNA damage, yet whether Sam68 regulates carcinogenesis through the regulation of NF-κB signaling is unknown.

In the submitted manuscript, Fu et al. showed that Sam68 regulates ADP ribose (PAR) polymer production and the PAR-dependent NF-κB transactivation of anti-apoptotic genes. As such, Sam68-deficient cells are hypersensitive to genotoxicity caused by DNA damaging agents. Importantly, this finding correlates with the high levels of Sam68 found in human colon cancer and with the results in which genetic deletion of Sam68 leads to decreased colon tumor burden in the Minapc mouse model.

Altogether, the paper is well written, the figures are clear and the organization is logical. Furthermore, the mechanistic results are relatively novel and strongly support their conclusions. Overall, this manuscript will be a strong contribution to *eLife*.

Essential revisions:

1) The authors claim that GFP-Sam68 expression in Sam68 KO cells reconstitutes CPT-induced IκBα degradation (Figure 1—figure supplement 1). The present data are not fully convincing. Further analyses, e.g. determination of p65 phosphorylation, NF-κB DNA binding activity or nuclear uptake are required to demonstrate that ectopic expression of Sam68 reconstitutes the PARP1/NF-κB axis in Sam68 KO cells.

2) In the first experiments (Figure 1) cells were treated with 10 µM CPT, which is a typical concentration for cell culture experiments. However, later on the authors increase the CPT concentration to 100 µM (Figure 1, Figure 2), which might generate misleading results. Thus, these experiments have to be repeated with 10 µM CPT treatment.

3) The analysis of HCT116 cells indicates constitutive PARP1 and NF-κB activation. However, it has been shown previously by others that although HCT116 cells display a certain basal NF-κB activity, NF-κB can be significantly induced by genotoxic stress. Thus, it would be important to analyze if loss of Sam68 or PARP1 inhibitor olaparib treatment affects stimulus dependent NF-κB activation and subsequent Bcl-XL expression. Since ATM is the second crucial component of DNA damage induced NF-κB activation, the authors should also analyze if Sam68 affects ATM S1981 phosphorylation.

4) Figure 7—figure supplement 1. The effective concentrations of PJ-34 for PAR inhibition (panel A) (2, 10 and 25 µM) and apoptosis induction (panel C) (25 and 50 µM) do not agree. The authors need to address this discrepancy.

5) Most of the conclusions are based on Western blots. Although some of them are quite clear, others are not. It will be important for the authors to quantify their results from the three independent repeats and show that their results are significant. Furthermore, it will be nice if some of the independent repeats are placed in the supplemental information. at least for the critical experiments. If the blots cannot be readily quantified (because they were developed on film and/or are overexposed), duplicates of all key experiments should be shown in the supplement. Also, blots should be cropped less closely.

Specific points:

1) In Figure 5, the authors compare colon tumor tissue with adjacent normal colon tissue, both from mice and patient material and analyze different features of PARP1/NF-κB signaling. Although the observations fit with previous findings, these data are only correlative, which the authors should acknowledge.

2) Ramakrishnan and Baltimore (2011) showed that NF-κB activation induced by TNF but not IL-1 is greatly reduced in Sam68 knockout cells (via receptor proximal interaction, Sam68 was proposed to promote RIPK1 and cIAP1 recruitment). Could a similar mechanism play a role in the cytoplasmic ATM-dependent cascade that is co-required for DNA damage induced NF-κB activation?

3) The effect of CPT treatment (10 µM, 2 h) on IκBα expression differs between Figure 1 (lane 1 and 2) and Figure 1 (lane 1 and 4). It has to be clarified if 10 µM CPT is sufficient to induce IκBα.

4) The immunoblot analysis of IKKγ (Figure 4) indicates that IKKγ is preferentially located in the nucleus, which is quite unusual. Is this also true for IKKα and IKKβ? Most likely, the blots are mixed up.

5) The authors could discuss that LRP16 has been described as a crucial regulator of DNA damage induced NF-κB activation, by binding to PARP1 and IKK (Wu et al. 2015). Could LRP16, as well as the Ku antigens, be mechanistically linked to Sam68?

6) Figure 2: The figure caption must have been switched (*-/-* versus *+/-* Sam68).

7) Figure 3: Why is the 'GFP only' signal around 27 kDa seen in all lanes?

8) Figure 7: Olaparib is normally used in concentrations between 1 and 10 µM. The authors should explain why they use 20 µM in their experiments.

9) The Introduction is quite convoluted. In particular the first paragraph. It will be important to write this section more clearly for readers that are not that familiar with the topic.

10) Why does the ΔC mutant interact with PARP1 if it is located in the cytoplasm?

11) The authors demonstrated that Sam68 is required for PARP1 activation and PAR production. Taking into account that Sam68 appears to have many functions and may be involved in a variety of cellular processes, it would be interesting to better understand how SAM68 contributes to PARP1 activation. Is Sam68 recruited to DNA lesions, together with PARP1? Does loss of Sam68 affect recruitment of PARP1 to DNA lesions? Although these revisions are not deemed essential, they would certainly flesh out the mechanism. The authors are therefore encouraged to add such data if they have them or if they are readily obtainable.

[Editors' note: further revisions were requested prior to acceptance, as described below.]

Thank you for resubmitting your work entitled "Sam68/KHDRBS1 is critical for colon tumorigenesis by regulating genotoxic stress-induced NF-κB activation" for further consideration at *eLife*. Your revised article has been favorably evaluated by Tony Hunter as the Senior Editor, a Reviewing Editor, and one reviewer.

The authors have performed a number of new analyses, satisfying most of the issues that were raised. However, new results that are now included raise problems that have to be solved in a second round of revision:

1) One control experiment that was asked for was to analyze ATM activation. They now show in Figure 6—figure supplement 1 that knockdown of Sam68 in HCT116 cells causes a significant attenuation of ATM phosphorylation. If Sam68 is needed for ATM activation, then this alone, without considering PARP1, could explain that Sam68 deletion blunts NF-κB activation by DNA damage.

2) In Figure 7—figure supplement 1 they show that the PARP1 inhibitor Olaparib strongly reduces ATM phosphorylation in HCT116 cells. This observation is completely unexpected. The Stilmann et al. paper (cited in the manuscript) shows in Supplemental Figure 7 that two different PARP1 inhibitors did not affect ATM phosphorylation and that ATM phosphorylation was not affected by PARP1 knockout (Figure 6, input). If the authors wish to maintain their claim that PARP1 activity is needed for ATM activation, they have to include further experiments, including the analysis of additional cell types. How would PARP1 control ATM activation? Thus, either additional experiments have to be performed or Figure 7—figure supplement 1 should be removed. The clarification of these points is important, since Olaparib has been approved as cancer drug. To my knowledge, it is not known that Olaparib blocks ATM activation.

3) The authors have shown no data to support their claim that IKKβ is degraded in the absence of Sam68 after CPT treatment. All they show is that its expression goes down. The reviewers did not notice this important point in the first round of review.

4) Size marker positions and the SUMOylated IKKγ signal. In Figure 1 this signal is slightly above 55 kDa (which is small considering the migration of unmodified IKKγ). In Figure 4 is at 70 kDa. Furthermore, in Figure 4—figure supplement 1, the PARP1 signal in the nuclear fraction is extremely faint.

[Editors' note: further revisions were requested prior to acceptance, as described below.]

Thank you for resubmitting your work entitled "Sam68/KHDRBS1 is critical for colon tumorigenesis by regulating genotoxic stress-induced NF-κB activation" for further consideration at *eLife*. Your revised article has been favorably evaluated by Tony Hunter as the Senior editor and the Reviewing editor.

The manuscript has been improved but there is still one remaining issue. In the second round of review, point 1, it was noted: "If Sam68 is needed for ATM activation, then this alone, without considering PARP1, could explain that Sam68 deletion blunts NF-κB activation by DNA damage." In other words, the authors had not formally addressed whether the effect of Sam68 on NF-κB goes via ATM or PARP1, and they did nothing in the revision to address this point experimentally. Given that PARP1 is required for NF-κB signaling and they show that Sam68 is upstream of PARP1, their interpretation is reasonable. However, they should discuss the possibility that alternatively, or in addition, Sam68 may also be operating via the ATM branch. This is particularly relevant given that the published literature (Stillman et al., 2009) reports that PARP1 activity is not required for ATM activation (see point 2 in re-review). Please add a few sentences to the Discussion to make readers aware of these issues.

---

## [Author Response]

*Essential revisions:*

1) The authors claim that GFP-Sam68 expression in Sam68 KO cells reconstitutes CPT-induced IκBα degradation (Figure 1—figure supplement 1). The present data are not fully convincing. Further analyses, e.g. determination of p65 phosphorylation, NF-κB DNA binding activity or nuclear uptake are required to demonstrate that ectopic expression of Sam68 reconstitutes the PARP1/NF-κB axis in Sam68 KO cells.

We have conducted the suggested experiments and our results demonstrate that ectopic expression of Sam68 in Sam68 KO cells reconstitutes the DNA damage-induced p65 phosphorylation and p65 nuclear translocation, thus further supporting the claim that supplementing Sam68 reconstitutes DNA damage-induced NF-κB activation signaling in Sam68 KO cells. These results have been included in the revised Figure 1—figure supplement 1.

2) In the first experiments (Figure 1) cells were treated with 10 µM CPT, which is a typical concentration for cell culture experiments. However, later on the authors increase the CPT concentration to 100 µM (Figure 1, Figure 2), which might generate misleading results. Thus, these experiments have to be repeated with 10 µM CPT treatment.

We agree with the reviewers that treatment with high concentration of CPT might generate misleading results. As suggested, we have repeated the identified experiments with 10 μM of CPT treatment. The results with 10 μM of CPT treatment are comparable to those with 100 μM of CPT treatment, therefore we have included the repeated results with 10 μM of CPT treatment in the revised Figure 1 and Figure 2.

*3) The analysis of HCT116 cells indicates constitutive PARP1 and NF-κB activation. However, it has been shown previously by others that although HCT116 cells display a certain basal NF-κB activity, NF-κB can be significantly induced by genotoxic stress. Thus, it would be important to analyze if loss of Sam68 or PARP1 inhibitor olaparib treatment affects stimulus dependent NF-κB activation and subsequent* Bcl-XL *expression. Since ATM is the second crucial component of DNA damage induced NF-κB activation, the authors should also analyze if Sam68 affects ATM S1981 phosphorylation*.

We appreciate that the reviewers pointed out that NF-κB can be significantly induced by genotoxic stress in HCT116 cells. Our additional experiments, as well as presented results, demonstrate that knockdown of Sam68 or PARP1 inhibitor Olaparib treatment attenuates genotoxic stress-induced NF-κB activation, as assayed by p65 phosphorylation (the revised Figure 6—figure supplement 1 and Figure 7—figure supplement 1), and subsequent Bcl-XL expression (Figure 6 and the revised Figure 7—figure supplement 1). Therefore, loss of Sam68 or PARP1 inhibitor Olaparib treatment affects both endogenous DNA damage and exogenous stimulus dependent NF-κB activation and subsequent Bcl-XL expression in HCT116 cells.

We have also analyzed whether Sam68 affects S1981 phosphorylation of ATM, the crucial component of DNA damage-induced NF-κB activation, in cellular response to genotoxic stress. Either knockdown of Sam68 or PARP1 inhibitor Olaparib treatment substantially attenuates DNA damage-induced ATM S1981 phosphorylation (the revised Figure 6—figure supplement 1 and Figure 7—figure supplement 1). Thus these data, together with our presented results, further support the crucial role of Sam68 in the DNA damage-triggered NF-κB activation signaling in the nucleus.

4) Figure 7—figure supplement 1. The effective concentrations of PJ-34 for PAR inhibition (panel A) (2, 10 and 25 µM) and apoptosis induction (panel C) (25 and 50 µM) do not agree. The authors need to address this discrepancy.

We apologize for the discrepancy, which might be caused by the misleading PAR immunoblot image with a very short exposure. We have rerun the samples and our results show that the effective concentrations of PJ-34 for PAR inhibition (panel A) correlate well with apoptosis induction in HCT116 cells (panel C, now the revised Figure 7—figure supplement 1).

5) Most of the conclusions are based on Western blots. Although some of them are quite clear, others are not. It will be important for the authors to quantify their results from the three independent repeats and show that their results are significant. Furthermore, it will be nice if some of the independent repeats are placed in the supplemental information. at least for the critical experiments. If the blots cannot be readily quantified (because they were developed on film and/or are overexposed), duplicates of all key experiments should be shown in the supplement. Also, blots should be cropped less closely.

As suggested by the reviewers, we have quantified most of our Western blot results from the three independent repeats, with statistical analyses, and included them in the revised Figure 1, Figure 2, Figure 4. We have also included the duplicates of certain key experiments in the supplement, including the revised Figure 2—figure supplement 1, Figure 2—figure supplement 4, Figure 3—figure supplement 1, and Figure 6—figure supplement 2.

*Specific points:*

1) In Figure 5, the authors compare colon tumor tissue with adjacent normal colon tissue, both from mice and patient material and analyze different features of PARP1/NF-κB signaling. Although the observations fit with previous findings, these data are only correlative, which the authors should acknowledge.

We appreciate that the reviewers pointed out that the PARP1/NF-κB signaling data in colon tumor tissue versus adjacent normal colon tissue are only correlative. The Results section on Figure 5 has been revised to acknowledge the point.

2) Ramakrishnan and Baltimore (2011) showed that NF-κB activation induced by TNF but not IL-1 is greatly reduced in Sam68 knockout cells (via receptor proximal interaction, Sam68 was proposed to promote RIPK1 and cIAP1 recruitment). Could a similar mechanism play a role in the cytoplasmic ATM-dependent cascade that is co-required for DNA damage induced NF-κB activation?

The reviewers’ interesting hypothesis stimulates more in-depth discussion about the distinct functions of Sam68 in multiple NF-κB activation signaling pathways. We showed that Sam68 interacts with PARP1 and IKKγ in the signaling complex after DNA damage; previous studies from the Miyamoto lab demonstrate that the translocation of IKKγ and ATM from the nucleus to the cytoplasm is essential for the cytoplasmic ATM-dependent cascade leading to NF-κB activation (Wu et al. Science 2006). Therefore, Sam68 could complex with the cytoplasmically translocated IKKγ-ATM complex and execute additional important functions for NF-κB activation and signaling in the cytoplasm. Based on our results from the subcellular fractionation experiments, we *did not* observe an obvious translocation of Sam68 from the nucleus to the cytoplasm during a 3-hour period following CPT treatment (Figure 1). Moreover, in our pilot experiments, we failed to detect the interaction between ATM and Sam68, when conducting the immunoprecipitation/immunoblot assays using the cytoplasmic subcellular fraction derived from CPT-treated cells (see the Figure below). These results indicate that Sam68 might not execute an additional function in the cytoplasm via participating in the cytoplasmic ATM-dependent cascade for DNA damage-induced NF-κB activation. Instead, as supported by our presented results, Sam68 plays an important role in the early signaling cascade in the nucleus to facilitate the NF-κB-activating signaling complex assembly. The distinct roles of Sam68 in the different subcellular signaling cascades for NF-κB activation induced by DNA damage versus TNF receptor engagement could indicate the existence of stimulus-dependent post-translational modifications. We have included these considerations in the Discussion.

Author response image 1.Whole cell lysates (WCL) from wild-type MEFs stimulated with 10 μM of CPT for indicated periods were immunoblotted (IB) directly or after subcellular fractionation and immunoprecipitation (IP) with Sam68 antibody for indicated proteins.**DOI:**
http://dx.doi.org/10.7554/eLife.15018.024

3) The effect of CPT treatment (10 µM, 2 h) on IκBα expression differs between Figure 1 (lane 1 and 2) and Figure 1 (lane 1 and 4). It has to be clarified if 10 µM CPT is sufficient to induce IκBα.

We appreciate the reviewers for pointing out the difference in the effect of CPT treatment (10 μM, 2h) on IκBα expression between Figure 1. Our repeated experimental results (the revised Figure 1) demonstrate that 10 μM of CPT treatment is sufficient to trigger IκBα degradation as well as additional signaling events that lead to NF-κB activation.

4) The immunoblot analysis of IKKγ (Figure 4) indicates that IKKγ is preferentially located in the nucleus, which is quite unusual. Is this also true for IKKα and IKKβ? Most likely, the blots are mixed up.

We are sorry for the misleading IKKγ image in our original version. The reviewers were correct that the majority of IKKγ, as well as IKKα and IKKβ, are located in the cytoplasm in mouse primary colon epithelial cells (CECs), like other cell types. It is noteworthy that a substantial amount of IKKγ is located in the nucleus, albeit not profound as that in the cytoplasm. Slightly different distribution patterns of certain proteins in the subcellular localizations have been noticed in multiple cell types; therefore, we believe that the observed subcellular localization of IKKγ in mouse CECs is cell context-dependent. Consistently, our repeated experiments demonstrate that genotoxic stress induced IKKγ translocation from the nucleus to the cytoplasm is profound in Sam68-sufficient CECs, in contrast to the significantly attenuated IKKγ cytoplasmic translocation in Sam68-deleted cells (the revised Figure 4), thus supporting the critical role of Sam68 in DNA damage-induced NF-κB activation signaling in the nucleus.

5) The authors could discuss that LRP16 has been described as a crucial regulator of DNA damage induced NF-κB activation, by binding to PARP1 and IKK (Wu et al. 2015). Could LRP16, as well as the Ku antigens, be mechanistically linked to Sam68?

As pointed out by the reviewers, Wu and colleagues recently reported that leukemia related protein 16 (LRP16) is a crucial regulator of DNA damage-induced NF-κB activation by binding to PARP1 and IKKγ (Wu et al., Nucleic Acids Res. 2015). The LRP16-PARP1 and LRP16-IKKγ interactions both *depend on* the PAR chain formation and the macro domain (which binds the terminal ADP-ribose of PAR) on the C-terminus of LRP16 is critical for such interactions, therefore the authors proposed that LRP16 functions downstream of PARP1-catalyzed PAR production and the affinity of LRP16 to PAR plays a critical role in facilitating the high-efficiency interactions of LRP16 with PARP1 and IKKγ to assemble the signaling complex that leads to NF-κB activation. In contrast, our results demonstrate that loss of Sam68 greatly impedes PARP1-catalyzed PAR production and the Sam68-PARP1 interaction is *not PAR-dependent* (see Figure 10), therefore Sam68 functions upstream of PAR and the interactions of LRP16 with PARP1 and IKKγ in DNA damage-induced NF-κB signaling.

Author response image 2.Wild-type MEFs, pretreated with DMSO or PJ-34 (10 μM) for 1h, were stimulated with 10 μM of CPT for indicated periods.Whole cell lysates (Input) were immunoblotted (IB) directly or after immunoprecipitation (IP) with PARP1 antibody for the indicated proteins.**DOI:**
http://dx.doi.org/10.7554/eLife.15018.025

Moreover, the role of LRP16 in DNA damage-induced NF-κB activation is dependent on the Ku70/Ku80 complex that binds specifically to double strand breaks (DSBs) (Wu et al., Nucleic Acids Res. 2015). Ku70 and Ku80 are known PARP1 substrates and PARP1 and PARP1-catalyzed PARylation are required to recruit and/or retain Ku70/Ku80 complex at DSBs (Coute et al., J Cell Biol. 2011). In contrast, our results demonstrate that DNA damage-induced PARylation of PARP1 and Ku70 is substantially attenuated in Sam68 KO MEFs (Figure 2—figure supplement 1), thus again supporting that Sam68 functions upstream of PARP1/PAR and Ku70/Ku80/LRP16 in DNA damage-induced NF-κB signaling.

As suggested, we have included this recent work on LRP16 in the revised Discussion.

6) Figure 2: The figure caption must have been switched (-/- versus +/- Sam68).

Sorry for the mistake – it has been corrected now.

7) Figure 3: Why is the 'GFP only' signal around 27 kDa seen in all lanes?

The signal around 25 kDa seen in all lanes was immunoglobin light chain recognized by the used GFP antibody. We have re-blotted the membrane with a different GFP antibody that exhibits cleaner and more specific GFP signal, and the improved image has been included in the revised Figure 3.

8) Figure 7: Olaparib is normally used in concentrations between 1 and 10 µM. The authors should explain why they use 20 µM in their experiments.

Besides the concentrations between 1 and 10 μM, 20 μM of Olaparib has also been used in some previous works, e.g. PNAS (2012) 109: 6590-6595 and Sci Rep. (2015) 5: 10129; we therefore used that dose in our original experiments. As suggested, we have repeated the experiments in Figure 7 with 10 μM of Olaparib, and the results are comparable to those using 20 μM treatment. Although 20 μM of Olaparib treatment does not cause misleading observations in our original experiments, it would be more appropriate to present the results generated from 10 μM of Olaparib treatment (please see the revised Figure 7).

9) The Introduction is quite convoluted. In particular the first paragraph. It will be important to write this section more clearly for readers that are not that familiar with the topic.

We appreciate the reviewers’ suggestion and have revised the Introduction to make it clearer for readers that are not that familiar with the topic.

10) Why does the ΔC mutant interact with PARP1 if it is located in the cytoplasm?

As illustrated by the images in intact cells, Sam68 (ΔC) mutant and PARP1 are preferentially located in the cytoplasm and the nucleus, respectively, and they do not overlap (interact). It is noteworthy that nuclear proteins (including PARP1) and cytoplasmic proteins (including Sam68 (ΔC) mutant) were lysed together and subjected to immunoprecipitation/immunoblot, when we carried out the structural functional analyses to map the essential domain(s) in Sam68 that confers the Sam68-PARP1 interaction. In the lysed cells without nuclear boundary, Sam68 (ΔC) mutant and PARP1 were therefore available to each other under this condition. Because Sam68 (ΔC) mutant contains the required domain(s) to bind PARP1, we observed the “apparent” Sam68 (ΔC)-PARP1 interaction in the immunoprecipitation/immunoblot assays. Of note, similar to the pulldown assays using recombinant proteins, our immunoprecipitation/immunoblot assays only suggest that Sam68 (ΔC) mutant with the required domain(s) harbors the capability to interact with PARP1. However, in the truly intact cells, as pointed by the reviewers, Sam68 (ΔC) mutant and PARP1 are preferentially localized in the cytoplasm and the nucleus, respectively, and they do not interact, therefore ectopic expression of Sam68 (ΔC) mutant fails to reconstitute DNA damage-induced IκBα degradation in Sam68 KO cells (Figure 1—figure supplement 1).

11) The authors demonstrated that Sam68 is required for PARP1 activation and PAR production. Taking into account that Sam68 appears to have many functions and may be involved in a variety of cellular processes, it would be interesting to better understand how SAM68 contributes to PARP1 activation. Is Sam68 recruited to DNA lesions, together with PARP1? Does loss of Sam68 affect recruitment of PARP1 to DNA lesions? Although these revisions are not deemed essential, they would certainly flesh out the mechanism. The authors are therefore encouraged to add such data if they have them or if they are readily obtainable.

We appreciate that the reviewers have suggested several very insightful experiments to further address the interesting question on how Sam68 contributes to PARP1 activation. While we have some preliminary results that could answer some of these questions and are actively following these directions, we feel that to fully answer these questions is out of the scope of our current study to illustrate a novel role of Sam68 in genotoxic stress-induced NF-κB activation and colon tumor survival. Once the essential role of Sam68 in early DNA damage signaling is established, we will be more than happy to fully elucidate the molecular mechanisms on how Sam68 contributes to PARP1 activation, globally and locally, in our follow-up studies.

[Editors' note: further revisions were requested prior to acceptance, as described below.]

*The authors have performed a number of new analyses, satisfying most of the issues that were raised. However, new results that are now included raise problems that have to be solved in a second round of revision:*

1) One control experiment that was asked for was to analyze ATM activation. They now show in Figure 6—figure supplement 1 that knockdown of Sam68 in HCT116 cells causes a significant attenuation of ATM phosphorylation. If Sam68 is needed for ATM activation, then this alone, without considering PARP1, could explain that Sam68 deletion blunts NF-κB activation by DNA damage.

As suggested, we have examined the impact of Sam68 knockout on DNA damage-induced ATM activation. Our results shown in Figure 11 demonstrate that Sam68 deletion substantially attenuates DNA damage-triggered ATM activation, as indicated by ATM Ser1981 phosphorylation, in MEFs and mouse primary thymocytes, which, together with our other data in the manuscript, further support the essential role of Sam68 in early DNA damage signaling. Of note, one previous study from the Poirier laboratory showed that ATM possesses PAR-binding motifs thus interacting with PARP1 in DNA damage response and that ATM activation depends on DNA damage-triggered PARP1 activation and subsequent PAR production (Haince et al. J Biol Chem. 2007 282: 16441-16453). We speculate that the attenuated ATM activation could result from the defective PARP1 activation and PAR synthesis in Sam68 knockout cells that we have demonstrated in our other results included in the manuscript, although we could not rule out the possibility that Sam68 may regulates ATM activation directly. Therefore these results also support our claim that Sam68 is a key NF-κB regulator in genotoxic stress-initiated signaling pathway and crucial for DNA damage-stimulated PAR production and the PAR-dependent NF-κB activation.

Author response image 3.(**A**) Wild-type (WT) and Sam68 knockout (KO) mouse embryonic fibroblasts (MEFs) were γ-irradiated (IR) at 4 Gy and whole cell lysates were derived at indicated time points post IR and immunoblotted (IB) for the indicated proteins, with β-actin as a loading control. (B) Primary thymocytes isolated from *Khdrbs1*^+/-^ and *Khdrbs1*^-/-^ mice were IR at 4 Gy and whole cell lysates were derived at indicated time points and IB for the indicated proteins, with β-actin as a loading control. p-ATM, Ser1981 phosphorylated ATM.**DOI:**
http://dx.doi.org/10.7554/eLife.15018.026

2) In Figure 7—figure supplement 1 they show that the PARP1 inhibitor Olaparib strongly reduces ATM phosphorylation in HCT116 cells. This observation is completely unexpected. The Stilmann et al. paper (cited in the manuscript) shows in Supplemental Figure 7 that two different PARP1 inhibitors did not affect ATM phosphorylation and that ATM phosphorylation was not affected by PARP1 knockout (Figure 6, input). If the authors wish to maintain their claim that PARP1 activity is needed for ATM activation, they have to include further experiments, including the analysis of additional cell types. How would PARP1 control ATM activation? Thus, either additional experiments have to be performed or Figure 7—figure supplement 1 should be removed. The clarification of these points is important, since Olaparib has been approved as cancer drug. To my knowledge, it is not known that Olaparib blocks ATM activation.

We agree with the reviewers that since Olaparib has been approved as a cancer drug, it is important to examine the available data carefully and a substantial amount of future studies are needed to better understand the effect of Olaparib on DNA damage signaling, particularly in regards to ATM activation. However, the relationship between PARP1 and ATM activation is not the major claim of our current work, we have taken the reviewers’ and editors’ suggestion and removed Figure 7—figure supplement 1 panel from the revised manuscript. While we may not fully agree with the reviewers’ interpretation of the data shown in Figure 6B and Supplementary Figure 7 in the Stillmann paper (Stillmann et al. Mol Cell 2009 36: 365-378), the suggested removal of the Figure 7—figure supplement 1 panel will not affect our major conclusions. The differences in our observations and the Stillmann paper could be caused by a variety of factors, such as exposure time of the western blot, the antibodies being used, and likely the difference in the dose of IR used. The 80 Gy dose used by the Stillmann paper is substantially higher than the dose we used (10 Gy), which may result in a saturated DNA damage signal (including the activation of ATM). This could account for the lack of observed difference in the phosphorylation of ATM in the WT versus PARP1^-/-^ MEFs (as well as the untreated versus PARP1 inhibitor-treated cells). We feel these points, combined with the previously mentioned evidence in the Haince paper which suggests that ATM activation is dependent on PAR production, are sufficient to support our data regarding the decrease in ATM activation following Olaparib treatment; however as this is not our main point and does not affect our overall conclusions, we have respectfully removed this data from the revised manuscript.

3) The authors have shown no data to support their claim that IKKΒ is degraded in the absence of Sam68 after CPT treatment. All they show is that its expression goes down. The reviewers did not notice this important point in the first round of review.

We assumed the concern is that we need to provide additional evidence further supporting our claim that IκBα is degraded in the presence of Sam68 after CPT treatment (as we do not currently show any data involving IKKΒ). Our additional experimental results show that CPT treatment induces significant decrease in the levels of IκBα in the DMSO vehicle control-pretreated WT MEFs, whereas the CPT-dependent reduction in IκBα levels is fully blocked by a pretreatment with MG132, a proteasome inhibitor. These data, consistent with numerous previous studies, strengthen our claim that IκBα is degraded after CPT treatment. We have included these results in the revised Figure 1—figure supplement 1.

*4) Size marker positions and the SUMOylated IKKγ signal. In Figure 1 this signal is slightly above 55 kDa (which is small considering the migration of unmodified IKKγ). In Figure 4 is at 70 kDa. Furthermore, in Figure 4—figure supplement 1, the PARP1 signal in the nuclear fraction is extremely faint.*

The separation time and gel percentages utilized for the experiments in Figure 1 and Figure 4 could result in the slight difference in the migration of SUMOylated IKKγ relative to the size markers. We have applied the same size marker (70 kDa) to better illustrate the positions of the SUMOylated IKKγ bands in the revised Figure 1 and Figure 4. As suggested, we have replaced the PARP1 blot with a better-quality image in the revised Figure 4—figure supplement 1.

[Editors' note: further revisions were requested prior to acceptance, as described below.]

The manuscript has been improved but there is still one remaining issue. In the second round of review, point 1, it was noted: "If Sam68 is needed for ATM activation, then this alone, without considering PARP1, could explain that Sam68 deletion blunts NF-κB activation by DNA damage." In other words, the authors had not formally addressed whether the effect of Sam68 on NF-κB goes via ATM or PARP1, and they did nothing in the revision to address this point experimentally. Given that PARP1 is required for NF-κB signaling and they show that Sam68 is upstream of PARP1, their interpretation is reasonable. However, they should discuss the possibility that alternatively, or in addition, Sam68 may also be operating via the ATM branch. This is particularly relevant given that the published literature (Stillman et al., 2009) reports that PARP1 activity is not required for ATM activation (see point 2 in re-review). Please add a few sentences to the Discussion to make readers aware of these issues.

As suggested, we have included the discussion about these important issues in the revised Discussion section (first paragraph).